# Probing the edge-related properties of atomically thin MoS$_2$ at nanoscale

Teng-Xiang Huang[1,4], Xin Cong[2,3,4], Si-Si Wu[1], Kai-Qiang Lin [1], Xu Yao[1], Yu-Han He[1], Jiang-Bin Wu[2], Yi-Fan Bao[1], Sheng-Chao Huang[1], Xiang Wang[1]*, Ping-Heng Tan [2,3]* & Bin Ren [1]*

Defects can induce drastic changes of the electronic properties of two-dimensional transition metal dichalcogenides and influence their applications. It is still a great challenge to characterize small defects and correlate their structures with properties. Here, we show that tip-enhanced Raman spectroscopy (TERS) can obtain distinctly different Raman features of edge defects in atomically thin MoS$_2$, which allows us to probe their unique electronic properties and identify defect types (e.g., armchair and zigzag edges) in ambient. We observed an edge-induced Raman peak (396 cm$^{-1}$) activated by the double resonance Raman scattering (DRRS) process and revealed electron–phonon interaction in edges. We further visualize the edge-induced band bending region by using this DRRS peak and electronic transition region using the electron density-sensitive Raman peak at 406 cm$^{-1}$. The power of TERS demonstrated in MoS$_2$ can also be extended to other 2D materials, which may guide the defect engineering for desired properties.

[1] State Key Laboratory of Physical Chemistry of Solid Surfaces, Collaborative Innovation Center of Chemistry for Energy Materials (iChEM), MOE Key Laboratory of Spectrochemical Analysis & Instrumentation, Department of Chemistry, College of Chemistry and Chemical Engineering, Xiamen University, Xiamen 361005, China. [2] State Key Laboratory of Superlattices and Microstructures, Institute of Semiconductors, Chinese Academy of Sciences, Beijing 100083, China. [3] Center of Materials Science and Optoelectronics Engineering & CAS Center of Excellence in Topological Quantum Computation, University of Chinese Academy of Sciences, Beijing 100049, China. [4]These authors contributed equally: Teng-Xiang Huang, Xin Cong. *email: wangxiang@xmu.edu.cn; phtan@semi.ac.cn; bren@xmu.edu.cn

Two-dimensional (2D) transition metal dichalcogenides (TMDCs) have emerged as promising materials with remarkable physicochemical properties[1–4]. Almost all TMDCs inevitably contain structure defects generated from the fabrication process. Though these defects are small (few nanometers and even single atom), they can break the lattice symmetry, create a quantum-confined environment, and modify the electronic band structures. These heterogeneities affect the electron excitation, electron scattering, and also energy transfer between phonons and electrons[5–7]. Therefore, defects introduce unique electronic properties in TMDCs and in turn dictate their electrical, optical, and chemical behaviors[5–7]. Moreover, defect effect leads to the band bending and the varied carrier density results in a gradual change of electronic properties of the nearby pristine materials. The formed electronic property variation regions between defect and pristine material have significant but different impacts on the electronic and chemical properties of 2D materials[6–8]. It was found that the electron transfer rate of graphene decreased in the lattice disordered region, whereas increased in the electronic transition region (ETR)[8]. Therefore, quantitative measurement of the lengths of these two regions may guide the effective defect engineering. The electronic properties of 2D materials can be impressively influenced by the edge defects[5,7], e.g., the zigzag edge is metallic, whereas the armchair edge is semiconducting for $MoS_2$[9]. The identification of defect types may in turn help correlate defects and their electronic properties, and guide the effective defect engineering.

Up to now, considerable efforts have been devoted to characterizing and understanding the electronic properties of defects. However, it is still a great challenge because defects are so small that their signals may be obscured by the strong signals of the surrounding pristine materials. Although scanning tunneling spectroscopy is able to characterize the electronic properties of defects at nanometer or even atomic scale[6,10,11], it has to work under ultrahigh vacuum and ultralow temperature, which is much different from the real application environment of TMDCs.

In this work, we employed tip-enhanced Raman spectroscopy (TERS), which can simultaneously obtain the morphologic and chemical fingerprint information with a nanometer spatial resolution and even single-molecule sensitivity assisted by the huge electromagnetic field enhancement of the plasmonic Au or Ag tip[12,13], to probe the edge-related lattice structure and corresponding electronic properties of atomically thin $MoS_2$. We showed that TERS can spatially obtain distinctly different Raman features of the edge defects in mono- and bilayer $MoS_2$, and investigate their unique lattice vibration and electronic properties.

We observed an edge-induced Raman peak (396 cm$^{-1}$) activated by the double resonance Raman scattering (DRRS) process, and revealed the interaction of electron and phonon, which was significantly affected by the electronic band structures at the edge.

We further visualize the defect-induced band-bending region of the conduction band at **K** and **Q** states which are involved in the DRRS process by using the DRRS-related Raman peak at 396 cm$^{-1}$ and the ETR using the electron density-sensitive Raman peak at 406 cm$^{-1}$. Notably, we spatially distinguished zigzag and armchair edges by using their spectral features, which can hardly be realized with conventional spectroscopic techniques. The power of TERS demonstrated in $MoS_2$ can also be extended to other TMDCs materials, which may provide a deep understanding of the fundamental electronic properties and guide the defect engineering for desired properties.

## Results

**Raman features of different 1D defects in 1 L and 2 L $MoS_2$.** The scheme to measure the TERS signals on edge defects in $MoS_2$ materials is shown in Fig. 1a. We first measured the far-field Raman spectra of the sample followed by topological imaging by an atomic force microscope (AFM) to identify the interested defects. Finally, we measured the TERS signal (in either single point or imaging manner). In this work, we used the mechanically exfoliated[14,15] monolayer (1 L) and bilayer (2 L) $MoS_2$ with different one-dimensional (1D) defects (1 L wrinkle, 1 L edge, 1–2 L step, and 2 L edge) transferred onto an atomically smooth Au substrate as the sample, and the AFM image of the sample was shown in Fig. 1b. The far-field Raman spectra on 1 and 2 L pristine $MoS_2$, and TERS spectra obtained on different types of defects are shown in Fig. 1c. The TERS signals from these defects are clearly different from the far-field signal of 1 and 2 L pristine $MoS_2$, and the different types of defects also show strikingly different Raman features in both the peak position and peak intensity. Most interestingly, the two peaks at 220 and 396 cm$^{-1}$ marked in the Raman spectra selectively present on the defect. Such fine features are not induced by TERS tips (as shown in the near-field spectra of the basal plane in Fig. 1c, upper two spectra) and will not be possible to observe without the help of the high spatial resolution of TERS.

The broad Raman peak at ~220 cm$^{-1}$ can only be observed in defects with dangling bonds (unsaturated coordination, such as edges and step), and absent in wrinkle and pristine $MoS_2$, as shown in Fig. 1c. This peak has been assigned to the defect-induced longitudinal acoustic phonons (LA mode) with a DRRS feature[5,7,16,17], similar to that of the D-band in graphene[18]. For

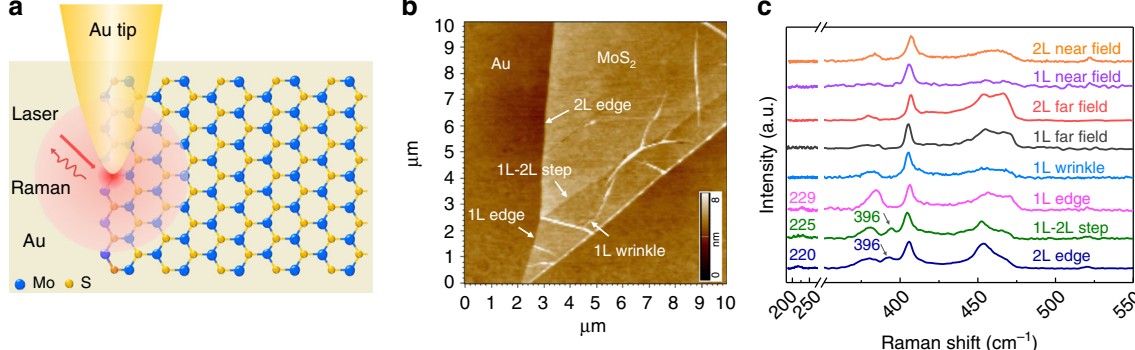

**Fig. 1** TERS study of edge defects in the atomically thin $MoS_2$. **a** Schematic of a TERS configuration using an Au-coated AFM tip and an atomically smooth Au film with monolayer (1 L) or bilayer (2 L) $MoS_2$ on the surface. **b** AFM image of the mechanically exfoliated $MoS_2$ with different types of 1D defect on an Au substrate. **c** Raman spectra of four 1D defects and the basal plane in $MoS_2$ marked in **b**, when the tip was approached and retracted. Note that these spectra are the near-field spectra and have been subtracted with the far-field signal, as well as the tip-enhanced photoluminescence background from $MoS_2$ and surface plasmon resonance (SPR) from the TERS tip. The intensity is normalized to the $A_{1g}$ peak for comparison.

the DRRS process of LA mode to occur, it requires one phonon and a defect for the momentum conservation[18–22]. The defect of the edge and step with dangling bonds will participate in the DRRS through the elastic scattering of the excited electron. However, the investigated wrinkle induces small lattice strain (≤0.4%; Supplementary Fig. 7), which exhibits little influence on lattice vibration[23]. Consequently, the wrinkle-assisted elastic scattering can be neglected, and the LA mode in wrinkle with a small curvature is absent according to the Raman selection rule.

In comparison, the 396 cm$^{-1}$ peak was observed in the defects in the multilayer MoS$_2$ (2 L edge and 1–2 L step in Fig. 1c) and was absent in the defects of monolayer MoS$_2$ (1 L edge and 1 L wrinkle in Fig. 1c). This peak was only explicitly observed and studied in the bulk MoS$_2$ at an ultralow temperature[24–26] and has not yet been observed experimentally in the defect at room temperature. It has been assigned to the combination of the LA and transverse acoustic (TA) phonons at the **M** point (LA(**M**) + TA(**M**)) of the Brillouin zone[27]. In addition, our wavelength-dependent measurement on the bulk MoS$_2$ at room temperature further reveals that this mode can only be activated under the resonance condition (Supplementary Fig. 15a; red and blue spectra) and the frequency of this peak shifts with the change of the laser excitation energy (Supplementary Fig. 15a; inset). The latter phenomenon is a signature of a DRRS, which requires photons with different energies to select electrons and phonons with different wave vectors in the Brillouin zone to satisfy the energy and momentum conservation[21]. This peak (396 cm$^{-1}$) is found to be related to the DRRS process between **K** and **Q** valleys and involves two phonons, i.e., LA(**M**) and TA(**M**) with wave vectors $q_M$ (Fig. 2a and Supplementary Fig. 15b). This DRRS process starts with an incoming photon (660 nm laser), whose energy matches the bandgap near **K** point, creating an electron–hole pair near **K** point. The electron can be excited by a resonant transition from the state in the valence band to the state in the conduction band. Then the excited electron can be resonantly scattered by emitting a phonon with wave vector $q_M$ to the electron state in the conduction band near **Q** point, when the energy of the **K** conduction band ($E_{CK}$) is larger than that of the **Q** conduction band ($E_{CQ}$). Afterward, the electron is inelastically scattered back to the virtual electron state near the **K** valley by emitting a second phonon with wave vector $-q_M$, where the electron–hole pair recombines and emits a photon with energy

$E_{laser} - \hbar\omega_{LA(M)+TA(M)}$ in valence band near **K** points. Therefore, the Raman intensity of this phonon can be greatly enhanced. Hence, when the energy of the **K** conduction band ($E_{CK}$) is higher than that of the **Q** conduction band ($E_{CQ}$; Fig. 2a), the probability of the DRRS process increases and thus enhances the corresponding Raman mode at 396 cm$^{-1}$. This peak is weak and even absent in the mono- and bilayer MoS$_2$ (Supplementary Fig. 15d) because $E_{CK}$ is smaller than $E_{CQ}$ (Supplementary Fig. 15b) in these two cases and the DRRS can hardly be activated. However, if there are defects in the mono- and bilayer MoS$_2$, they will lead to bending of the electronic band structure of MoS$_2$, resulting in a considerable change of $E_{CK}$ and $E_{CQ}$ (Fig. 2b, see Supplementary Note 3.3 for details about the electronic band structures of nanoribbon with different widths). As a result, the DRRS can be activated and the intensity of this peak can be greatly enhanced in the 1–2 L step and 2 L edge, since $E_{CK}$ becomes larger than $E_{CQ}$ (Fig. 2b; right two panels). Whereas, the possibility of DRRS is low in the 1 L wrinkle and 1 L edge due to the smaller $E_{CK}$ compared to $E_{CQ}$ (Fig. 2b; left two panels). This DRRS-involved mode opens up an opportunity to further study the intervalley scattering of electrons by acoustic phonons in the defect. It should be pointed out that it is almost impossible to observe this peak in the defect without the high spatial resolution provided by TERS.

**Nanoscale characterization of the edge in the bilayer MoS$_2$.** With the observed distinctive spectral features related to the defects, we further explored the different Raman features at the edge and basal plane of MoS$_2$. Figure 3a shows the topographic height profile (left panel) and the corresponding color-coded intensity map of the line-trace TERS image (right panel) of the bilayer edge marked in Supplementary Fig. 16b, inset. The larger height of the bilayer MoS$_2$ (~1.8 nm) compared with the typical bilayer thickness (1.4–1.6 nm)[28,29] is the result of a thin water layer existing between MoS$_2$ and the substrate[30–32] (Supplementary Figs. 1 and 2), which can block the doping effect from the Au substrate (Supplementary Note 1). The line-trace TERS image of the edge matches well with the height profile. When the tip was positioned on the Au substrate, only the far-field Raman signal of the pristine bilayer MoS$_2$ was observed. When the tip was moved gradually from the Au substrate to MoS$_2$, both the TERS spectral background contributed by tip-enhanced photoluminescence (TEPL) of the bilayer MoS$_2$ and the Raman signal are enhanced

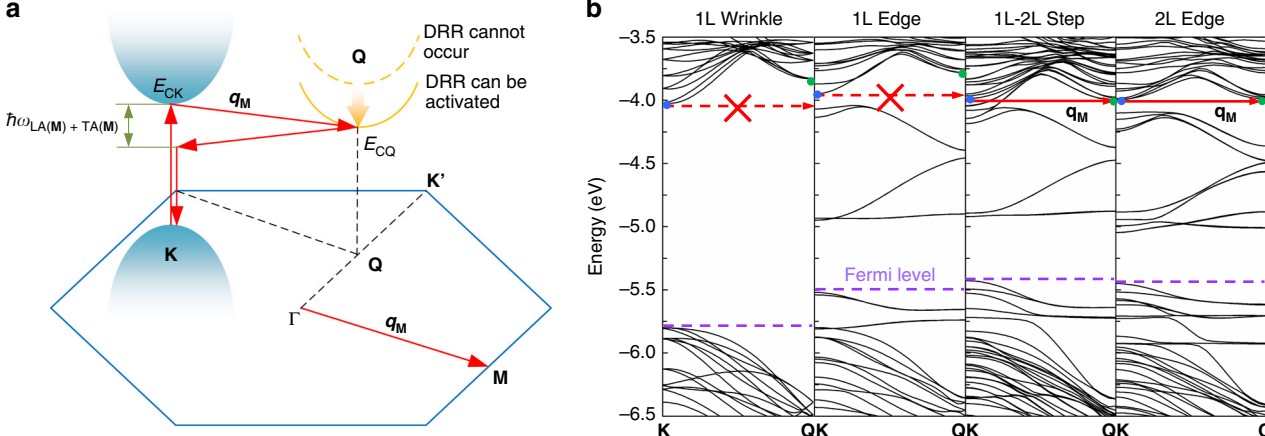

**Fig. 2** DRRS in the defect. **a** Representation of DRRS process of the LA(**M**) + TA(**M**) mode. **b** Electronic band structures of monolayer wrinkle (monolayer nanoribbon with 0.4% strain (Supplementary Figs. 6 and 7), for more details see Supplementary Note 3), monolayer armchair nanoribbon (1 L edge), monolayer armchair nanoribbon on the monolayer basal plane (1–2 L step), and bilayer armchair nanoribbon (2 L edge). Blue and green dots are **K** and **Q** points of conduction bands, respectively. The solid horizontal red arrows indicate the DRRS can occur and the dash arrows cannot occur. Note that the width armchair nanoribbon employed here is ~2.4 nm. See Supplementary Note 3.3 for details about the electronic band structures of nanoribbon with different widths (1–3 nm).

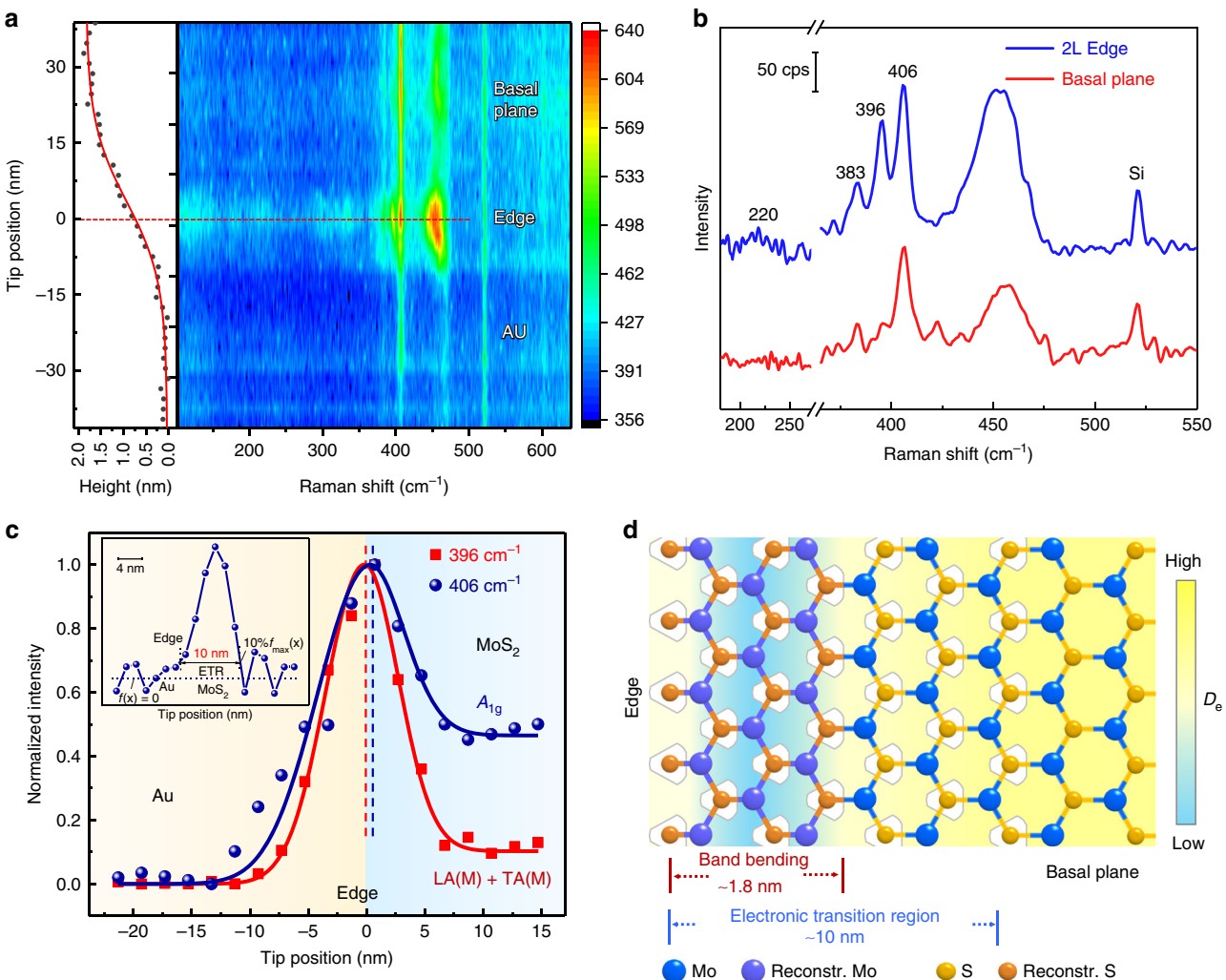

**Fig. 3** TERS characterization of a bilayer edge in MoS₂. **a** TERS line image across the edge of a bilayer MoS₂. Left panel: topographic height profile of the edge of bilayer MoS₂ marked in Supplementary Fig. 16b inset. Right panel: a line-trace TERS image of the edge. Note that these are original TERS experimental spectra without any processing. See Supplementary Fig. 16a for more data sets for the corresponding line-trace TERS spectra. **b** Two typical TERS spectra of a bilayer MoS₂ at the basal plane and the edge. **c** Plots of normalized intensities of two TERS peaks (396 and 406 cm⁻¹) with the tip position. The solid lines are the fitted results. Note that these spectra are the pure near-field signals and have been subtracted with the far-field signals. Inset is the enhanced TERS intensity profile of the ETR in the 2 L edge after the deconvolution of the EM field the intensity distribution (see Supplementary Note 2 for more details). **d** Schematic diagram of band reconstruction and ETR of MoS₂ near the edge. $D_e$ is the electron density.

and reach the maximum values at the edge. Interestingly, the TEPL spectra can be deconvoluted into a short wavelength part contributed by the neutral electron–hole pair ($A$-exciton) and a long wavelength part by the charged complex ($A^-$-trion; Supplementary Fig. 16c, d), and $A$ exciton PL dominates in the TEPL spectra at both edge and basal plane MoS₂. It has been reported that the carrier density can be estimated from the exciton and trion PL spectral weight[10,11,33]. We observed an increased $A$ integrated intensity and a decreased $A^-$ integrated intensity near the edge site, indicating that part of $A^-$ trions have lost electrons and transformed into $A$ excitons. Therefore, electron density near the edge is lower than that of the basal plane. A depletion of electrons weakens the charge screening effect, leading to a higher intensity of all the Raman modes at the edge[34,35] (Supplementary Fig. 17 shows that as the potential shifts positively, the intensity of all the Raman modes increases). Similarly, we can conclude that the other three types of 1D defects also have a lower electron density than the basal plane from the line-trace TERS image (Supplementary Fig. 18).

To clearly identify the difference in the spectral features at the edge and basal plane, we present two representative Raman spectra in Fig. 3b. It is clear that distinctly different Raman features, including the two new Raman peaks at 220 and 396 cm⁻¹, show up at the edge and disappear on the basal plane, indicating a significant difference in physicochemical properties (such as the lattice structure and electronic property) between the edge and basal plane[10,11,36]. As a result, transitional space forms between the edge and pristine MoS₂. Therefore, a precise measurement of this space is vital to the understanding of defect and to the defect engineering. As discussed above, the intensity of LA(**M**) + TA(**M**) mode is sensitive to $E_{CK} - E_{CQ}$. Thus, the intensity variation region of LA(**M**) + TA(**M**) mode at the edge can reflect the band-bending region of DRRS involved states (i.e., **K** and **Q** states in the conduction band; see Supplementary Note 3.3 for more details). We plot its intensity as a function of tip position as the red curve in Fig. 3c. The intensity becomes higher when the tip was moved closer to the edge, indicating a stronger DRRS-involved band bending at the edge. The full width at half maximum (FWHM) of

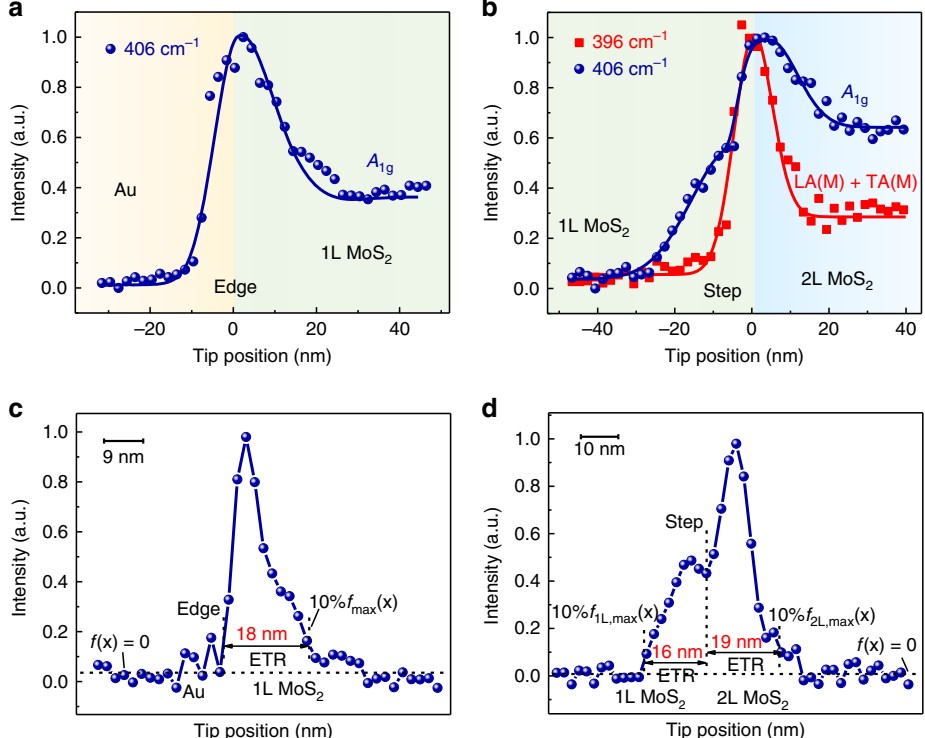

**Fig. 4** ETRs induced by 1 L edge and 1–2 L step. **a**, **b** Plots of normalized intensities of TERS peaks with the tip positions across the 1 L edge (**a**) and 1–2 L step (**b**). The solid lines are the fitted results. See Supplementary Note 2 for details about the data fitting. Note that these spectra are the pure near-field signals and have been subtracted with the far-field signals. Supplementary Fig. 20b, cshows more data sets for the corresponding line-trace TERS spectra. Enhanced TERS intensity profiles of the ETR in 1 L edge (**c**) and 1–2 L step (**d**) after the deconvolution of the EM field intensity distribution (see Supplementary Note 2 for more details).

the 396 cm$^{-1}$ intensity profile is 7 nm (Supplementary Fig. 5, and see Supplementary Note 2 for more details). It should be pointed out that the measured TERS intensity profile is a convolution of the DRRS-involved band-bending region ($d_{LA(M)+TA(M)}$) and the spatial resolution of TERS. Considering the typical spatial resolution of AFM-based TERS obtained in our lab of ~7 nm[12], which is equal to the experimentally measured TERS intensity profile, we expect the length of the DRRS-involved band-bending region to be much smaller than 7 nm. Indeed, theoretical calculation reveals that $d_{LA(M)+TA(M)}$ of bilayer $MoS_2$ induced by the edge only involves several lattice with a length of ~1.8 nm (shown in Supplementary Fig. 12; see Supplementary Note 3 and Supplementary Figs. 9–11 for more details). This DRRS-involved mode sheds light on the influence of defect on the band structure and energy bending for the involved electronic states.

Different from the 396 cm$^{-1}$ peak, the $A_{1g}$ peak (406 cm$^{-1}$) is sensitive to the electron density and its intensity becomes stronger when the tip is moved closer to the edge (Fig. 3c). Here, we neglect the phonon confinement at the edge because the phonon confinement region is small and it would decrease the intensity of $A_{1g}$ peak[17,35,37,38]. The stronger signal at the edge is attributed to a lower electron density[35], which is also supported by a lower TEPL intensity of the $A^-$ trion (Supplementary Fig. 16c, d). Two factors may account for the different electron densities at the edge and basal plane. First, the Fermi level at the edge is higher than that on the basal plane (Supplementary Fig. 13b), leading to a charge transfer from the edge to the basal plane. Second, the active edge site can be easily chemisorbed by $O_2$ or $H_2O$ in air (Supplementary Fig. 16e), which leads to the p-type doping induced by the oxygen species and a decreased electron density[33,34,39]. From this $A_{1g}$ intensity profile, we further obtained an ~10 nm ETR from the edge to the basal plane after

deconvoluting the spatial distribution of the electromagnetic field (Supplementary Figs. 3 and 4)[10,11]. Very interestingly, different from the maximum intensity of $LA(\mathbf{M}) + TA(\mathbf{M})$ mode (the largest DRRS scattering) located at the edge site, the maximum intensity of $A_{1g}$ mode (the lowest electron density) appears inside the pristine $MoS_2$ and is located just a few nanometers away from the edge (Fig. 3c inset and Supplementary Fig. 19). The reason for this is unclear yet at this moment, which may come from the confined edge state (a few lattice spacings)[40] (Supplementary Fig. 8e) and the depletion of the electron by the chemisorbed oxygen species near the edge rather than at the edge.

We further used TERS to investigate ETRs induced by the monolayer edge and the step between monolayer and bilayer (1–2 L) $MoS_2$. As shown in Fig. 4a, b, no 396 cm$^{-1}$ signal can be detected at the monolayer edge, and the 396 cm$^{-1}$ intensity fluctuates in a narrow region (FWHM ~6.9 nm) across the 1–2 L step, indicating a high TERS spatial resolution in these measurements. Similar to the bilayer edge, the electron density becomes smaller when the tip is moved closer to the edge and the step. The lowest electron density (i.e., the maximum $A_{1g}$ intensity) appears inside the pristine $MoS_2$, and is located in just a few nanometers away from the edge (Fig. 4c) and the step (Fig. 4d). In addition, we can obtain a larger ETR of ~18 nm induced by the monolayer edge (Fig. 4c) than that of 10 nm induced by the bilayer edge (Fig. 3c inset). This may be a result of a larger Fermi energy difference between the monolayer edge and the basal plane (Supplementary Fig. 13b), as well as a deeper oxygen p-type doping due to the higher chemical activity compared with that of 2 L $MoS_2$[41]. Interestingly, we observed two obviously different ETRs near the 1–2 L step (Fig. 4d). The step is doped by oxygen and acts as a narrow quantum wire[40], and thus induces ETRs near the step in both monolayer (~16 nm)

and bilayer (~19 nm). The above study clearly reveals the defect-induced ETR of $MoS_2$ in the real application environment, which guides better engineering of the defect density to achieve the desired properties.

**Determination of different edge structures.** We have demonstrated the incredible ability of TERS to spatially investigate the edge-related lattice structure and electronic properties of atomically thin $MoS_2$ flakes previously. It would be very interesting to determine the periodic arrangement of different atoms (zigzag or armchair) of $MoS_2$ as it has a significant impact on the band gap, magnetic, optical, and electronic properties[5,42–44]. In the graphene case, the $D$ peak has been used to identify different edge structures because it is inactive at the zigzag edge but active at the armchair edge[42,45]. Unfortunately, no such clear conclusion has been made to allow the identification of different edge structures of TMDCs, to the best of our knowledge. Interestingly, during our line-trace TERS study, in addition to the increased intensity of the $A_{1g}$ mode (406 cm$^{-1}$), we often found two opposite changing trends of its peak position when the tip moved from the basal plane to the edge: downshift (Fig. 5a, b; left panels) and upshift (Fig. 5a, b; right panels). The $A_{1g}$ frequency may change with the relaxation of momentum conservation induced by the phonon confinement effect and the weak electron–phonon coupling due to the reduced electron density at the edge. However, these two

effects should only lead to the small upshift of the $A_{1g}$ frequency[17,35,37,38], which cannot fully illustrate the two opposite directions of the frequency shift at the edge. We then consider the influence of local strains introduced by different edge structures on the $A_{1g}$ frequency with density functional theory (DFT) calculation. The calculation result given in Fig. 5c shows that the frequency of the $A_{1g}$ mode at the armchair edge would upshift compared with that on the basal plane, whereas there is a significant downshift in the $A_{1g}$ frequency at the zigzag edge for both S-terminated and Mo-terminated. As a result, the $A_{1g}$ frequency downshifts at the zigzag edge compared with the basal plane (Fig. 5a). In addition, the FWHM of the $A_{1g}$ peak increases at both the zigzag and armchair edges (Supplementary Fig. 21) due to the disorder of the structure at the edge.

To further verify the conclusion drawn above, we mechanically exfoliated 1 L $MoS_2$ with different edge angles (60° and 90°) on the Au substrate, as indicated by the dotted lines in Fig. 5d. These special angles would provide the structure information of the two adjacent edges[42,46,47]. For an angle of 60° (Fig. 5e), both edges have the same structure (either zigzag or armchair). The frequency shift direction of the $A_{1g}$ mode at both edges should be either downshift or upshift. On the other hand, for an angle of 90° (Fig. 5e), the two edges have different structures, i.e., one armchair and the other zigzag (Mo- or S-terminated). The $A_{1g}$ frequency at these two edges should shift to different directions,

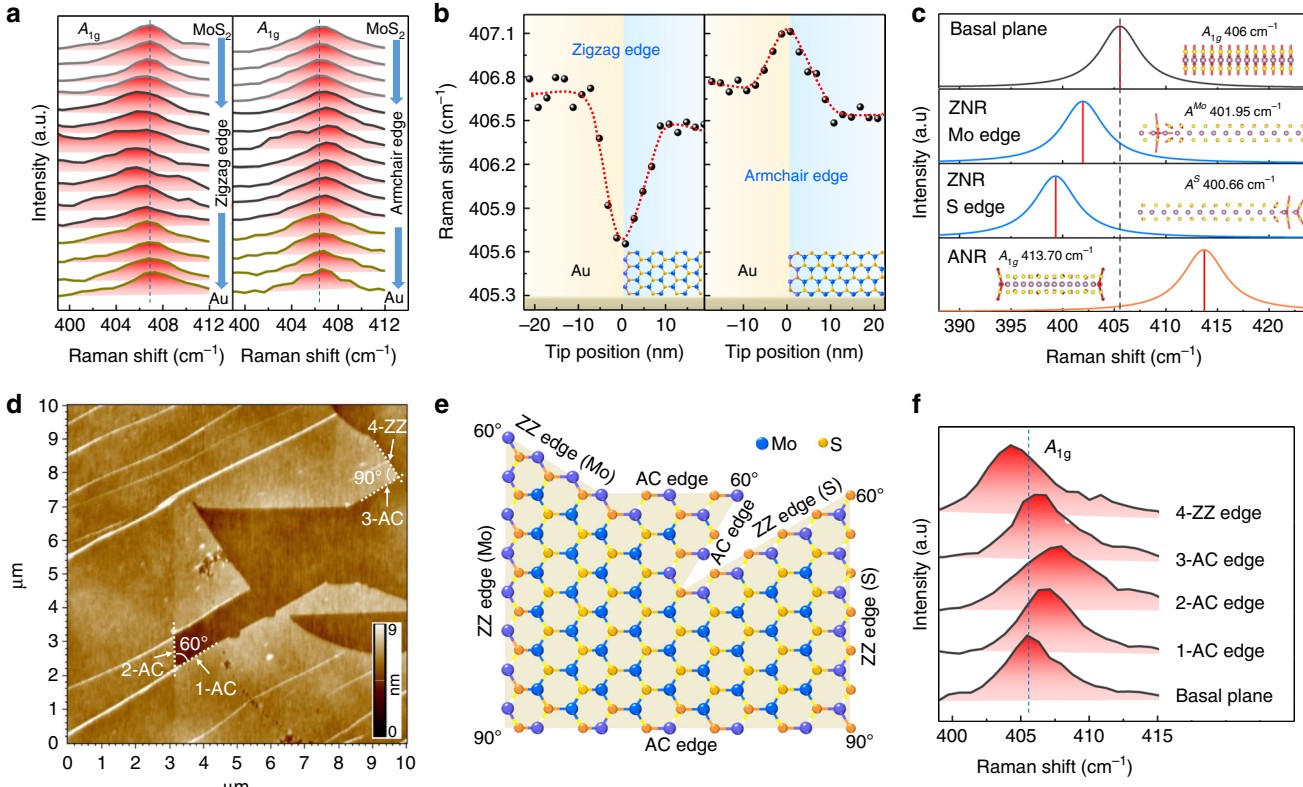

**Fig. 5** Effect of the edge structure on the peak position of the Raman $A_{1g}$ mode. (**a**) Typical line-trace TERS spectra of the zigzag edge (left panel) and armchair edge (right panel) in the spectral range of the $A_{1g}$ mode. Note that the spectra have been normalized with the intensity of $A_{1g}$ peak for a better comparison. (**b**) Plots of peak position with the tip position. The corresponding line-trace TERS spectra are shown in Supplementary Fig. 16a and Supplementary Fig. 20a, respectively. The dash red lines are guides for the eye. (**c**) Calculated Raman spectra and lattice vibration of the basal plane, zigzag nanoribbon (ZNR, with a width of 3.59 nm) localized at the Mo and S edges, and armchair nanoribbon (ANR, with a width of 2.05 nm). The envelope of each Raman spectrum is evaluated by smearing the peaks with half width 5 cm$^{-1}$. Note that the displacements of $A_{1g}$ or $A_{1g}$-like mode with out-of-plane vibration are side viewed. See Supplementary Note 3.4 for details. (**d**) AFM image of a mechanically exfoliated 1 L $MoS_2$ with different edge angles on an Au substrate. (**e**) Illustration of the relationship between angles and edge structures of zigzag (ZZ) and armchair (AC) in 2 H $MoS_2$. (**f**) TERS spectra of four edges in the spectral range of the $A_{1g}$ mode marked in **d**. Note that the spectra have been normalized with the intensity of $A_{1g}$ peak for a better comparison. The corresponding line-trace TERS spectra of these edges are shown in Supplementary Fig. 22.

i.e., one upshift and the other downshift. Therefore, for a sample containing both 60° and 90° edge angles, the four edges should contain either three AC and one ZZ or one AC and three ZZ. Indeed, on TERS spectra of these edge (Fig. 5f), we detected three upshifted edge and one downshifted edge, which can be convincingly assigned to AC and ZZ edges following the conclusion of our rigorous theoretical calculation, respectively. Therefore, we demonstrate here that TERS can conveniently identify the edge type of TMDC materials under the ambient condition, which will be very important for the practical application of TMDCs.

## Discussion

We have spatially resolved the unique electronic property of the defects in atomically thin $MoS_2$, as well as the influence of defects on the nearby pristine material, by using TERS with 7 nm spatial resolution. We observed a Raman peak at 396 cm$^{-1}$ in the defect of bilayer $MoS_2$ that is rather difficult to observe and obscured by conventional Raman microscopy. We assigned it to the phonon vibration activated by two-phonon DRR intervalley scattering, and subsequently unveiled the interaction of electron and phonon during the electron excitation and scattering, which was significantly affected by the electronic band structures in different defects. We successfully determined the lengths of edge-induced band-bending region (~1.8 nm) of DRRS involved conduction band at $K$ and $Q$ states (~1.8 nm) and the ETR (~18 nm near 1 L edge, ~10 nm near 2 L edge, and ~16 nm (monolayer) and ~19 nm (bilayer) near the step) in the real space by taking the advantage of the high spatial resolution (7 nm) of TERS. We further demonstrated that TERS can be effectively used to identify edge structures using the frequency shift direction of the $A_{1g}$ mode at 406 cm$^{-1}$, with an upshift at the armchair edge and downshift at the zigzag edge. This work provides a general approach to investigate the defect-related electronic properties in 2D materials by TERS, which may assist in revealing the structure–function relationship of the defect and subsequently guide the effective defect engineering and promote the applications of TMDC materials.

## Methods

**Sample preparation**. An atomically smooth Au substrate was prepared following a template-stripping method proposed by Hegner et al.[48]. First, Au (Zhongnuo Advanced Material Technology Co., Ltd. 99.999%) was vapor-coated onto a Si wafer in a vacuum chamber at a pressure of $5 \times 10^{-7}$ torr at a slow deposition rate of 0.35 nm s$^{-1}$ to form a layer with a thickness of 200 nm. Afterward, small glass slides were glued to the Au film using UV adhesive (Norland Optical Adhesive, Norland Products Inc., NJ). The glue was then exposed to UV light for 10 min; the slide can be saved for use. Before use, the glass slide is to be peeled off to reveal the atomically smooth Au surface.

$MoS_2$ samples were obtained by mechanical exfoliation[14,15]. In brief, freshly cleaved $MoS_2$ was fabricated on the Nitto tape and then pasted onto the viscoelastic polydimethylsiloxane stamp (Gel-park, WF-20-X4) to produce plenty of thin $MoS_2$ flake. The stamp was inspected under the optical microscope to select the interested $MoS_2$ flakes and finally pressed against the exposed Au surface to transfer $MoS_2$. No doping effect from the Au substrate on $MoS_2$ was observed (see Supplementary Note 1 for more details).

**AFM and TERS measurements**. An upright Raman microscope (NTEGRA Spectra, NT-MDT) was employed for the TERS measurement[49]. The system is equipped with a confocal laser microscope, an AFM, and a white-light video microscope for rough observation/alignment of the sample and tip. A $100 \times$ long working-distance objective with a numerical aperture of 0.7 was used for both excitation and collection of the backscattered light from the sample. A 660 nm laser was used for TERS measurements in a p-polarized configuration. Au-coated AFM tips used in TERS experiments were produced by electrodeposition of Au onto silicon AFM tips (VIT_P/IR, NT-MDT)[49]. The radii of the as-prepared tips were ~65 nm. The sample topography was measured using a tapping mode AFM and TERS spectra were obtained using a contact mode AFM. The doping effect induced by the Au tip on the edge and the basal plane is the same, which will not influence the conclusion (see Supplementary Note 1 for more details). In the TERS line-trace imaging experiment, the tip was scanned at a velocity of 2 nm s$^{-1}$ and TERS spectra

were acquired simultaneously. Note that all the TERS data are the pure near-field spectra and have been subtracted from the far-field signal otherwise mentioned.

TEPL measurements were conducted on the same microscope (NTEGRA Spectra, NT-MDT) combined with a scanning tunneling microscope. The Ag metallic tip and the p-polarized laser beam (with a wavelength of 632.8 nm) were used. The constant current mode (300 pA tunneling current and 600 mV tip to sample bias) was employed to maintain the same electron tunneling to the basal plane and defects of $MoS_2$.

Raman measurements of bulk $MoS_2$ excited by different laser lines were conducted on the same Raman microscope (NTEGRA Spectra, NT-MDT) as mentioned above. The p-polarized laser beams (with wavelengths of 594, 632.8, 660, and 691 nm) were used for investigations.

**Theoretical simulations**. The DFT calculations were performed using the Vienna Ab Initio Simulation Package (VASP)[50]. The electron–ion interaction was described by the Projector Augmented Wave (PAW) pseudopotentials[51] and the plane–wave basis set with a kinetic energy cutoff of 500 eV. The exchange–correlation function is described by Perderw, Burke, and Ernzerhof (PBE) version of the generalized gradient approximation (GGA)[52]. The conjugated gradient method was performed for geometry optimization. The convergence condition for the energy was $10^{-8}$ eV. The influence of edge defect on pristine electronic and vibrational properties is estimated by zigzag and armchair nanoribbon with different widths (1–3 nm). The details of employed structures and Raman intensity calculations are presented in Supplementary Note 3. Unit cell and nanoribbon structures were relaxed until the force on each atom was less than $10^{-2}$ eV Å$^{-1}$. The Monkhorst–Pack special k-point meshes of the unit cell and nanoribbon structures were $11 \times 11 \times 1$ and $1 \times 5 \times 1$, respectively. The phonon dispersion of nanoribbons was calculated by using $1 \times 3 \times 1$ supercell and based on DFT implemented in VASP software in connection with the Phonopy software[53].

## Data availability
The data that support the findings of this study are available from the corresponding author upon reasonable request.

## Code availability
The code that used in this study are available from the corresponding author upon reasonable request.

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

## Acknowledgements

The authors acknowledge the final supports from MOST of China (2016YFA0200601 and 2016YFA0301204), NSFC (21633005, 21790354, 21503181, 21711530704, 21621091, 11874350, 11474277, and 11434010), Natural Science Foundation of Fujian Province (2016J05046), and China Postdoctoral Science Foundation (2017M622062).

## Author contributions

B.R., P.-H.T. and X. W. supervised the project. T.-X.H., X.W. and B.R. conceived the ideas. T.-X.H., S.-S.W., K.-Q.L. and Y.-F.B. performed the experiments. X.Y. fabricated the Au substrate. X.C., J.-B.W. and P.-H.T. performed the DFT calculations. Y.-H.H. and S.-C.H. helped with experiments and analysis. All authors contributed to data interpretation and writing of the manuscript.

## Competing interests

The authors declare no competing interests.
