## [Peer Review File · Nature Communications]

Reviewers' Comments:

Reviewer #1:

Remarks to the Author:

The authors present a comprehensive study of edges and defects of atomically thin MoS₂ flakes by Tip-enhanced Raman spectroscopy with 7 nm of spatial resolution. They detect, at the edge of the MoS₂ flake, a forbidden Raman mode for the monolayer at 396 cm⁻¹ which they assigned to the combination of the LA and TA phonons at the M point of the Brillouin zone. This Raman mode, allowed only in bulk samples and in resonant conditions (double resonant Raman scattering, DRRS), is detected at the edge of MoS₂ with a spatial extent of 7 nm. The authors assigned this mode to the lattice reconstruction which, from their calculations, is predicted to be 0.64 nm large. This is consistent with their spatial resolution limit of 7 nm. On the other hand, the A_{1g} mode at 406cm⁻¹, which is sensitive to electron density, is also detected at the edge but with a larger extent. The authors claim that the variation in intensity of the A_{1g} mode is due to a depletion region coming from the p-doping induced by oxygen at the edge. They also found that the frequency of the A_{1g} phonon changes by probing the edge and it can shift in both directions: up and down in energy depending on the edge type. By the comparison of the data with calculations, the authors attribute the frequency downshift to zigzag edge and the frequency upshift to armchair edge.

I think that the manuscript shows impressive results and it is compliant with the criteria of publication of Nature Communication because: the data is technically sound, the paper provides strong evidence for its conclusions, the results are novel and the manuscript is important to scientists in the specific field. However, I would invite the authors to revise their manuscript to address specific concerns which I describe below in order of importance.

1. In the second paragraph "Nanoscale characterization of the edge in bilayer MoS₂", pag. 9. From "Different from the 396 cm⁻¹ peak..." the authors discuss the intensity variations of the A_{1g} phonon and of the photoluminescence (A-exciton and A-trion) as a function of spatial variations of electron density and they attribute the stronger signal at the edge to a low electron density due to p-doping from adsorbed oxygen. Here the doping due to the charge transfer from the gold substrate is not mentioned. The contact between MoS₂ and gold can also change the Fermi level [Scientific Reports v. 4, Art. Num. 5575, DOI: 10.1038/srep05575]. Despite I agree that in a bilayer this effect is less important than in a monolayer, the doping induced by the contact with the gold (or silver) tip should be at least considered or briefly mentioned. If the authors neglected the doping by the substrate because it is constant, they should state it clearly.

In the section "Methods", the authors claim that they used contact mode AFM, but in S.I. they describe this experiment as STM-TERS using Ag tips. In this case, the applied voltage should be also specified. It is not clear in which way this experiment has been carried out, but the utilized method could strongly influence the conclusion. For instance, regarding the data in Fig.4 c, an additional p-doping induced by the tip could upshift the phonon frequencies. This effect could explain the discrepancy between the experimental and calculated frequency of the A_{1g} mode. In general, I found the paragraph starting from "Different from the 396 cm⁻¹ peak..." until "to achieve desired properties" a bit confusing with some repeated concept not in a logical order. The sentence "The stronger signal at the edge attributed to a low electron density as a result of p-type doping" should be better connected with the "decrease in intensity expected from phonon confinement" and justified or supported with citation (as previously done in pag. 7 with "charge screening" and refs. 28, 30).

I would suggest rephrasing and checking the grammar of this paragraph. (e.g. "preformed" in place of "performed", "is" missing, etc..)

2. In pag 4, Fig. 1, The authors wrote "Note that the background of the spectra has been subtracted" but in pag 8 Fig 3 they wrote "Plots of normalized intensities of two TERS peaks (396 and 406 cm⁻¹) with the tip position after the subtraction of far field signals". And similar sentences are written in S.I.

The disentanglement of the local information, or pure near field (PNF), from the mixed signal is an important issue in TERS. A method to extract the local information is usually to subtract the

spectrum acquired with the tip retracted to the spectrum acquired with the tip engaged (PNF = NF – FF). From the sentences written in the manuscript is not clear if the authors made this operation or if they just subtracted the broad plasmonic background which usually appears in TERS spectra. In the first case one can obtain the real local information and appreciate peak shifts which could be hidden by the far field, in the second case one just subtract the plasmonic quadratic background. I agree that this subtraction is not necessary for peaks which appear only when the tip is approached, but in my opinion the authors should be clearer about this point and clearly state it, for instance, in the section “Methods”.

3. The LA(M)+TA(M) double resonant Raman mode is well explained in the manuscript and supported by the data in S.I., but for the reader is hard to understand quickly when they refer to the canonical DRRS or to the near-field double resonant Raman scattering in TERS. Because of this peak appears in monolayer MoS₂ only by approaching the tip, I would suggest to discern the two different processes by calling the second as tip-enhanced double resonant Raman scattering, i.e. with the acronyms TE-DRRS.

4. In the pag 2, 7th row after the abstract the authors wrote “involved in the energy transfer process between different quantum states such as quasi-particles (e.g. excitons) and elementary particles (e.g. phonons).”

I have to admit that reading this it was quite shocking for me: phonons are labelled quasi-particles by every textbook. As far as I know, phonons are quasi-particles, bosons, and excitons are quasi-particles too, but fermions. I would suggest reviewing this sentence because phonons are not elementary particles. [Nature Physics volume 12, pages 1085–1089 (2016), <https://www.nature.com/articles/nphys3977>]

5. I would suggest adding at the stack plot in Fig. S7a in S.I. a vertical scale in nanometers and adding a line in Fig S7b (the AFM image) which indicates where the line-trace has been performed, maybe a zoom in is necessary. If another high resolution AFM image which zoom in on the edge has been acquired, I would suggest adding it and also showing the height profile of the morphology of the edge. In general, the height profile of the edge is important because if the edge is for some reason detached from the substrate, the doping changes due to the lack of charge transfer (as I described in point 1), like for instance demonstrated for wrinkles in graphene [Nano Lett., 2015, 15 (2), pp 857–863, DOI: 10.1021/nl503460p].

6. The authors at pag 3 state “We observed a new defect-induced Raman peak (396 cm⁻¹)...” and successively they assign it to the combination mode LA(M)+TA(M). In my opinion they could name it, if they want it, and if no one else did it already. It could be also easier to address to E mode (edge mode) in the text for instance. This decision is really up to the authors, is not important for the acceptance of the manuscript.

7. The section “Methods” appears twice, once before the references with only a short sentence and then after in the middle of the references, between ref 37 and 38, with also Acknowledgements etc..

Be careful for the final print layout.

Dr. Emanuele Poliani

Reviewer #2:

Remarks to the Author:

Dear Editor,

the paper report measurements which might be interesting. Unfortunately the interpretation is very lacking and the computational results used to support the scenario is far from being acceptable.

I do not recommend this paper for publication.

Concerning the interpretation of the various peaks, overall the authors do not discuss the possible alternative scenarios but make a series of claim without any real factual support. Moreover, the use of the calculations to support the interpretation of the various peaks is far from being convincing. This is a pity given that certain measurements appear as interesting.

Certain points should be addressed before any kind of publication:

1) Concerning the band at 220 cm^{-1} . The authors attribute it to a double resonant mechanism. One of the arguments is that it is not observed in the wrinkle defect (Fig.1) because "the wrinkle ... does not fulfill the momentum conservation".

This is bad english. I guess that the authors want to say that the wrinkle-defect does not break the translational invariance symmetry which allows the to relax the momentum conservation allowing double resonant processes to happen (as in the case of the graphite/graphene D defect). The problem is that this is not the case: a wrinkle is expected to break translational invariance symmetry. The fact that the 220 peak is observed in the presence of dangling bonds and not for the wrinkle is an indication that this peak is possibly associated with a localized vibration?

2) Fig.2b reports the electronic bands calculated for a nanoribbon of finite width. In this kind of plot (well understood and commonly used in literature) one observes an important number of bands that correspond to "bulk" electronic bands + certain bands associated to states localized at the edge.

For a finite-width ribbon, the "bulk" bands are affected by quantum confinement effects and, for example, the apparent electronic gap will be different from that of the bulk and will change in a visible way by increasing the ribbon width. The relative position of the conduction and valence bands in Figure 2 is thus strongly dependent on the width of the ribbon. This kind of study makes sense only if it is done on several different ribbons with varying width.

This point is relevant since the bands of Figure 2 are used in the interpretation of the measurements.

2) If the authors want to claim that the band gap is affected by electronic states at the edges they should prove the present of surface electronic states at the minimum/maximum of the conduction/valence bands. This cannot be evinced from Figure 2 (actually it looks quite unlikely from that figure).

3) Figure2. In panel b there are four electronic bands. It is relatively clear how the bands labeled "1L" and "2L" have been obtained. It is not clear how "wrinkle" and "1L-2L step" calculations were done. I could not find an understandable description neither in the text nor in the Supplemental Information.

4) Line 178/179. "The Fermi level at the edge is higher than that of the basal plane, leading to charge transfer...". It is not clear to me how the change in Fermi level observed in the calculations (done in vacuum) could possibly be compared to a depletion possibly present in the measurements (done on flakes lying on a surface). How do the work functions of the structures of Fig. S5b? How are they affected by the presence of a substrate?

5) Figure 4, panel c. Concerning the calculations. How are these numbers calculated? The authors simply provide a couple of points in the figure without describing how they were obtained.

The authors simulated some ribbons and then they calculate the phonons. The phonons that you can obtain with a ribbon calculations have the same problems of interpretation of the electronic bands discussed above. Are we talking about vibrations localized at the edge? How can you decouple the two edges of the ribbon by using ribbons with such a small width? Are we talking about vibrations of the bulk? How are these vibrations affected by the confinement of the vibration within the width of the ribbon? In their structures the authors have tens of vibrational modes with overlapping frequencies. What is the criterion chosen to plot the points reported in Figure 4C? This kind of study makes sense only if ribbons with different widths are studied as a function of the width. Using just one ribbon can be misleading.

6) Calculations on the ribbons. The details of the calculations are not enough. What is the lattice spacing? The structure of the ribbons are described in Figure S10. What is the criterium to distinguish among "reconstructed bonds" and "pristine bonds"? If you repeat the same calculation (geometry optimization) with a wider ribbon, how are the bond lengths affected? (this kind of calculations is not particularly heavy, but it is part of the standard tests done with these kind of problems). The materials reported in the manuscript is not enough to judge about the reliability of these calculations.

Minor points:

7) Figure 2. In the first lines of the caption the authors talk about strain and then refer to Figure S6 of the Supplementary Info for more details. Figure S6 does not seem to be related to this discussion.

8) Throughout the paper the authors use the word "reconstruction" to refer (I guess) to atomic relaxation present near the edges of the structure. In surface science, the term "reconstruction" has a specific meaning which does not correspond to this. To avoid confusion they should avoid the use of the term "reconstruction".

Sincerely,

Reviewer #3:

Remarks to the Author:

In this work, the authors demonstrate the capabilities of TERS to identify spectral features that exist only near defects. The experimental work seems to have been carried out with care, and the data should attract some interest from researchers in the related fields. However, there are a few issues that prevents this reviewer from recommending publication of this work in Nature Communications.

First of all, the importance or impact of the work is questionable. It turned out that the main focus of this work is the edge of 2-layer MoS₂. The authors explain that the special band reconstruction near edges of 2L MoS₂ allowed the defect-related Raman features to show up. This means that this method is useless for monolayer MoS₂ which is far more interesting and important for other thicknesses. In this sense this work would have a very limited impact which is very disappointing.

Then there are issues of interpretation. Some of the interpretations are not based on experimental evidence but on inference based on calculations. Since there are experimental means to verify the claims, the authors should strive to back up their claims with experimental evidence.

1) In page 10, the authors claim that the maximum intensity position of the A1g mode is located a few nanometers away from the edge. However, the two vertical lines in Fig. 3c are separated by less than 1 nm, much smaller than the spatial resolution of TERS. This small difference might be caused by the asymmetric profile of the A1g intensity (blue curve).

2) The analysis of the zigzag vs armchair edges is not fully substantiated by experimental data. It seems that the identification of the edge type is based on DFT calculations only. The authors should find ways to back up their claims with experimental data such as TEM, STEM, STM, SHG, etc. There are many ways to determine the edge types (approximately). Another way is to compare the results with samples grown by CVD. In CVD samples, the edge types are often determined from the shape of the platelets.

Also, some of the explanations are not correct or incomplete.

1) On page 4, the authors claim, "the wrinkle will only result in lattice strain rather than atoms with dangling bonds and does not fulfill the momentum conservation." Any defect that destroys the translational symmetry of the lattice would result in relaxation of the momentum conservation. There is no difference in the dangling bonds and local strain in this regard. The authors should find some other plausible explanation to explain the absence of the LA feature at wrinkles.

2) The explanation for the LA(M)+TA(M) mode sounds plausible. Then, why the LA(M)+LA(M), in other words, the 2LA(M) mode is not enhanced at the edges? One would think that the latter is more likely excitation than the LA+TA scattering.

3) On the bottom of page 5, the authors explain the reconstruction of the electronic band near the edge defects. However, the band picture is not strictly valid in the presence of defects and the momentum conservation would be significantly relaxed, especially when the reconstruction is limited to the region of only a few nanometers in width.

4) I do not understand the following phrase in page 9: "... depletion region between the metallic edge and the semiconducting basal plane." The edges are more intrinsic than the basal plane because the p-doping depletes the excess electrons that are present in the basal plane. What is the meaning of 'metallic edge'? How can one see an exciton in a metallic material?

Some additional comments:

1) It would be nice to include TERS (not far field) spectra of pristine 1L and 2L basal planes in Fig. 1c.

2) It is well known that there are very high densities of S vacancies even in exfoliated MoS₂. It would be nice if the authors can isolate single S-vacancy with TERS and see if any spectral feature can be correlated.

Reviewers' comments:

Reviewer #1 (Remarks to the Author):

The authors present a comprehensive study of edges and defects of atomically thin MoS₂ flakes by Tip-enhanced Raman spectroscopy with 7 nm of spatial resolution. They detect, at the edge of the MoS₂ flake, a forbidden Raman mode for the monolayer at 396 cm⁻¹ which they assigned to the combination of the LA and TA phonons at the M point of the Brillouin zone. This Raman mode, allowed only in bulk samples and in resonant conditions (double resonant Raman scattering, DRRS), is detected at the edge of MoS₂ with a spatial extent of 7 nm. The authors assigned this mode to the lattice reconstruction which, from their calculations, is predicted to be 0.64 nm large. This is consistent with their spatial resolution limit of 7 nm. On the other hand, the A_{1g} mode at 406 cm⁻¹, which is sensitive to electron density, is also detected at the edge but with a larger extent. The authors claim that the variation in intensity of the A_{1g} mode is due to a depletion region coming from the p-doping induced by oxygen at the edge. They also found that the frequency of the A_{1g} phonon changes by probing the edge and it can shift in both directions: up and down in energy depending on the edge type. By the comparison of the data with calculations, the authors attribute the frequency downshift to zigzag edge and the frequency upshift to armchair edge.

I think that the manuscript shows impressive results and it is compliant with the criteria of publication of Nature Communication because: the data is technically sound, the paper provides strong evidence for its conclusions, the results are novel and the manuscript is important to scientists in the specific field. However, I would invite the authors to revise their manuscript to address specific concerns which I describe below in order of importance.

1. In the second paragraph “Nanoscale characterization of the edge in bilayer MoS₂”, page 9. From “Different from the 396 cm⁻¹ peak...” the authors discuss the intensity variations of the A_{1g} phonon and of the photoluminescence (A-exciton and A-trion) as a function of spatial variations of electron density and they attribute the stronger signal at the edge to a low electron density due to p-doping from adsorbed oxygen. Here the doping due to the charge transfer from the gold substrate is not mentioned. The contact between MoS₂ and gold can also change the Fermi level [Scientific Reports v. 4, Art. Num. 5575, DOI: 10.1038/srep05575]. Despite I agree that in a bilayer this effect is less important than in a monolayer, the doping induced by the contact with the gold (or silver) tip should be at least considered or briefly mentioned. If the authors neglected the doping by the substrate because it is constant, they should state it clearly.

Response1-1: We agree with the reviewer’s opinion that one may need to consider the doping effect. However, in our experiment, we did not find the doping effect induced by the substrate as discussed below. The Au tip did have a similar doping effect on the edge and the basal plane, which will not change the final conclusion. To address this point, we added a new section into the supplementary information:

Section 1. Doping effect from the Au substrate and Au AFM-TERS tip

Au substrates and Au tips were employed in this work. However, the doping effect of the substrate can be neglected. From the AFM height curve (Fig. 3a) and AFM images (Fig. R1a-d), we can find that a thin ice-like water layer (0.3~0.6 nm) exists between MoS₂ and the Au film, which can

block the doping effect from the Au substrate. Furthermore, as shown in Fig. R1e, the A_{1g} peak of monolayer MoS_2 on Au shows almost the same frequency as that on silica, indicating that no doping occurred between MoS_2 and Au. Only after the removal of water, the Au substrate could directly interact with MoS_2 , leading to the decrease of the PL intensity (Fig. R1f) and the upshift of the A_{1g} peak (Fig. R1g).

In the case of the tip, although there is still a small gap (~ 0.5 nm) existing between the tip and MoS_2 in the contact mode, we did find the charge transfer between tips and MoS_2 due to different work functions of tip materials (Au or Ag) and MoS_2 (Φ_{Ag} (4.3 eV) $<$ Φ_{MoS_2} (4.7 eV) $<$ Φ_{Au} (5.1 eV), Nanoscale 2016, 8, 10564) and the strong surface plasmon resonance at the tip apex. As shown in Fig. R2, when an Au tip approached MoS_2 , the electron is transferred from MoS_2 to the Au tip, leading to the p-doping of MoS_2 and the upshift of the A_{1g} peak. Conversely, when an Ag tip approached MoS_2 , the electron is transferred from the Ag tip to MoS_2 , leading to the n-doping of MoS_2 and the downshift of the A_{1g} peak. When the tips were coated with a thin silica layer, no doping effect was observed. Although charger transfer between the tip and MoS_2 existed in the TERS measurement, it will not influence the final conclusion since the tip did have a similar doping effect on the edge and the basal plane.

For clarity, we also revised the method part in the main text as follows:

- (1) In the sample preparation part, we add one sentence at the end of the paragraph: No doping effect from the Au substrate on MoS_2 was observed (see Supplementary Section 1 for more details).
- (2) In the AFM and TERS measurements part, we also make a statement in the first paragraph: The doping effect induced by Au tip on the edge and the basal plane is the same, which will not influence the conclusion (see Supplementary Section 1 for more details).

Figure R1. The presence of the water layer between the monolayer MoS_2 and Au film prevents the doping effect of Au substrate on MoS_2 . (a-d) AFM images of monolayer MoS_2 on

the Au film before and after annealing at 80 °C for a different time. (e) Raman spectra of monolayer MoS₂ on the Au film and silica substrate before annealing. The excitation laser wavelength was 532 nm and the laser power was 0.1 mW, and the acquisition time was 10 s. Photoluminescence (f) and Raman (g) spectra of monolayer MoS₂ on the Au film before and after annealing at 80 °C for a different time. The excitation laser wavelength in PL measurements was 632.8 nm with a laser power of 0.03 mW and the acquisition time was 1 s. The excitation laser wavelength in Raman measurements was 532 nm with a laser power of 0.3 mW and the acquisition time was 30 s.

Figure R2. Demonstration of the charge transfer between TERS tips and MoS₂. (a) Peak positions of the A_{1g} mode after the approaching of different tips. (b) Schematic of the charge transfer between the tip and MoS₂. The excitation laser was 632.8 nm.

In the section “Methods”, the authors claim that they used contact mode AFM, but in S.I. they describe this experiment as STM-TERS using Ag tips. In this case, the applied voltage should be also specified. It is not clear in which way this experiment has been carried out, but the utilized method could strongly influence the conclusion. For instance, regarding the data in Fig.4 c, an additional p-doping induced by the tip could upshift the phonon frequencies. This effect could explain the discrepancy between the experimental and calculated frequency of the A_{1g} mode.

Response1-2: We are sorry that we did not describe these two measurements in a sufficiently clear way in the previous version. We used the AFM contact mode and Au-coated AFM tips for all the TERS measurements, whereas we used STM mode and Ag tips only for the tip-enhanced photoluminescence (TEPL) measurements (Supplementary Fig. S16c and d) for the following two reasons. First, the AFM feedback laser and the surface plasmon resonance (SPR) of the rough gold coating layer on the AFM tip will affect the PL spectra of MoS₂. Second, the far-field PL signal of MoS₂ is so strong that we need to use Ag tips with a stronger SPR enhancement than Au tips to obtain a high near-field/far-field PL contrast. However, the influence of STM tips on the TEPL measurement can be neglected since we employed the constant current mode (300 pA tunneling current and 600 mV tip to sample bias), and thus the electron tunneling from the tip to the basal plane and defect is constant. For clarity, we add a description of experimental detail of the TEPL measurement in the method part as follows:

“Tip-enhanced photoluminescence (TEPL) measurements were conducted on the same microscope (NTEGRA Spectra, NT-MDT) combined with a scanning tunneling microscope (STM). The Ag metallic tip and the *p*-polarized laser beam (with a wavelength of 632.8 nm) were used. The constant current mode (300 pA tunneling current and 600 mV tip to sample bias) was employed to maintain the same electron tunneling to the basal plane and defects of MoS₂.”

In general, I found the paragraph starting from “Different from the 396 cm⁻¹ peak...” until “to achieve desired properties” a bit confusing with some repeated concept not in a logical order. The sentence “The stronger signal at the edge attributed to a low electron density as a result of p-type doping” should be better connected with the “decrease in intensity expected from phonon confinement” and justified or supported with citation (as previously done in page 7 with “charge screening” and refs. 28, 30). I would suggest rephrasing and checking the grammar of this paragraph. (e.g. “preformed” in place of “performed”, “is” missing, etc..)

Response1-3: We thank the reviewer for the kind suggestion and we have revised these two paragraphs as follows (page 7, second paragraph):

“Different from the 396 cm⁻¹ peak, the A_{1g} peak (406 cm⁻¹) is sensitive to the electron density and its intensity becomes stronger when the tip is moved closer to the edge (Fig. 3c). Here, we neglect the phonon confinement at the edge because the phonon confinement region is small and it would decrease the intensity of A_{1g} peak^{17, 36, 37}. The stronger signal at the edge is attributed to a lower electron density, which is also supported by a lower TEPL intensity of the A⁻ trion (Supplementary Fig. S16c,d). Two factors may account for the different electron densities at the edge and basal plane. First, the Fermi level at the edge is higher than that at the basal plane (Supplementary Fig. S13b), leading to a charge transfer from the edge to the basal plane. Second, the active edge site can be easily chemisorbed by O₂ or H₂O in air (Supplementary Fig. S16e), which leads to the p-type doping induced by the oxygen species and a decreased electron density^{33, 38, 39}. From this A_{1g} intensity profile, we further obtained a ~10 nm electronic transition region from the edge to the basal plane after deconvoluting the spatial distribution of the electromagnetic field (see Supplementary Section 2 for more details). Very interestingly, different from the maximum intensity of LA(M)+TA(M) mode (the largest DRRS scattering) located at the edge site, the maximum intensity of A_{1g} mode (the lowest electron density) appears inside the pristine MoS₂ and is located in just a few nanometers away from the edge (Fig. 3c inset and Supplementary Fig. S19). The reason is unclear yet at the moment, which may come from the confined edge state (a few lattice spacings)³⁸ (Supplementary Fig. S8e) and the depletion of the electron by the chemisorbed oxygen species near the edge rather than at the edge.”

2. In pag 4, Fig. 1, The authors wrote “Note that the background of the spectra has been subtracted” but in pag 8 Fig 3 they wrote “Plots of normalized intensities of two TERS peaks (396 and 406 cm-1) with the tip position after the subtraction of far field signals”. And similar sentences are written in S.I. The disentanglement of the local information, or pure near field (PNF), from the mixed signal is an important issue in TERS. A method to extract the local information is usually to subtract the spectrum acquired with the tip retracted to the spectrum acquired with the tip engaged (PNF = NF – FF). From the sentences written in the manuscript is not clear if the authors made this operation or if they just subtracted the broad plasmonic background which usually appears in TERS spectra. In the first case one can obtain the real local information and appreciate peak shifts which could be hidden by the far field, in the second case one just subtract the plasmonic quadratic background. I agree that this subtraction is not necessary for peaks which appear only when the tip is approached, but in my opinion the authors should be clearer about this point and clearly state it, for instance, in the section “Methods”.

Response: Sorry for the confusion. In this manuscript, spectra shown in Supplementary Fig. S16a,

and Supplementary Fig. S20 are the original TERS spectra without any data processing. The spectra shown in Fig. 1c have been subtracted with the far field signal as well as the near field background from TEPL of MoS₂ and SPR of Au tip. All the other TERS spectra have been subtracted only with the far field signals. For clarity, we rephrased the related part as follows:

(1) We have clearly stated it in the third paragraph of Methods as follows: **Note that all the TERS data are the near field spectra and have been subtracted with the far field signal otherwise mentioned.**

(2) We added a description in the caption of Figure 1: **Note that these spectra are the near field spectra and have been subtracted with the far field signal as well as the tip-enhanced photoluminescence background from MoS₂ and SPR from the TERS tip.**

(3) We added a description in the captions of Supplementary Fig. S16a and Supplementary Fig. S20: **Note that these spectra are the original TERS signals without any data processing.**

3. The LA(M)+TA(M) double resonant Raman mode is well explained in the manuscript and supported by the data in S.I., but for the reader is hard to understand quickly when they refer to the canonical DRRS or to the near-field double resonant Raman scattering in TERS. Because of this peak appears in monolayer MoS₂ only by approaching the tip, I would suggest to discern the two different processes by calling the second as tip-enhanced double resonant Raman scattering, i.e. with the acronyms TE-DRRS.

Response: We thank the reviewer for the kind suggestion. However, TERS did not provide special enhancement for the DRRS mode. It can enhance the Raman modes with the matched selection rule. The reason that we can detect the DRRS in the edge by TERS is that TERS provides a high spatial resolution to manifest the Raman signal of small features that are emerged in the far field signal of MoS₂ basal plane. To avoid the confusion, we add two TERS spectra of the monolayer and bilayer basal plane into Fig. 1c to indicate no impact from the TERS tip on the DRRS process.

4. In the page 2, 7th row after the abstract the authors wrote “involved in the energy transfer process between different quantum states such as quasi-particles (e.g. excitons) and elementary particles (e.g. phonons).” I have to admit that reading this it was quite shocking for me: phonons are labelled quasi-particles by every textbook. As far as I know, phonons are quasi-particles, bosons, and excitons are quasi-particles too, but fermions. I would suggest reviewing this sentence because phonons are not elementary particles. [Nature Physics volume 12, pages 1085–1089 (2016), <https://www.nature.com/articles/nphys3977>]

Response: Thanks for pointing out this conceptual problem. Following the reviewer’s suggestion, we have modified the sentence to:

“These heterogeneities affect the electron excitation and electron scattering, and also energy transfer between phonons and electrons.”

5. I would suggest adding at the stack plot in Supplementary Fig. S7a in S.I. a vertical scale in nanometers and adding a line in Fig S7b (the AFM image) which indicates where the line-trace has been performed, maybe a zoom in is necessary. If another high resolution AFM image which zoom in on the edge has been acquired, I would suggest adding it and also showing the height profile of the morphology of the edge. In general, the height profile of the edge is important because if the edge is for some reason detached from the substrate, the doping changes due to the

lack of charge transfer (as I described in point 1), like for instance demonstrated for wrinkles in graphene [Nano Lett., 2015, 15 (2), pp 857–863, DOI: 10.1021/nl503460p].

Response: We thank the reviewer for the kind suggestion and have modified the two figures. We have added a vertical topographic height profile of the edge of bilayer MoS₂ marked in Supplementary Fig. S16b inset where the TERS line-trace performed, as shown in Fig. R3a. We also added a zoom-in AFM image of the bilayer edge in Supplementary Fig. S16b inset, as shown in Fig. R3b. It shows that the edge and the basal plane have the same height. As we mentioned in question 1 that there is a thin water layer between MoS₂ and the substrate. Thus, the doping effect from the substrate can be neglected in our experiment.

Figure R3 Nanoscale characterization of bilayer edge in MoS₂. (a) Left panel: topographic height profile of the edge of bilayer MoS₂ marked in Fig. R3b inset. Right panel: the corresponding line-trace TERS spectra of the edge. Scan rate: 2 nm/step. The tip was scanned at a rate of 2 nm s⁻¹ and TERS spectra were acquired simultaneously with a laser power of 0.2 mW and an acquisition time of 0.4 s. (b) AFM image of the bilayer MoS₂ on an Au substrate. Inset: the zoom-in AFM image at the bilayer edge.

6. The authors at page 3 state “We observed a new defect-induced Raman peak (396 cm⁻¹)...” and successively they assign it to the combination mode LA(M)+TA(M). In my opinion they could name it, if they want it, and if no one else did it already. It could be also easier to address to E mode (edge mode) in the text for instance. This decision is really up to the authors, is not important for the acceptance of the manuscript.

Response: Indeed, the combination mode LA(M)+TA(M) can be observed not only near the edge of 2L MoS₂, but also in pristine NL MoS₂ with N>2. Furthermore, LA(M)+TA(M) mode can also be observed in the basal plane of 1L and 2L MoS₂ under a large strain. Consequently, it is not appropriate to define LA(M)+TA(M) as E mode (edge mode), and it is hard to present an exact name to emphasize its feature.

7. The section “Methods” appears twice, once before the references with only a short sentence and then after in the middle of the references, between ref 37 and 38, with also Acknowledgements etc..

Be careful for the final print layout.

Response: Thanks for pointing the problem and we removed all the duplications.

Reviewer #2 (Remarks to the Author):

Dear Editor,

The paper report measurements which might be interesting. Unfortunately, the interpretation is very lacking and the computational results used to support the scenario is far from being acceptable. I do not recommend this paper for publication. Concerning the interpretation of the various peaks, overall the authors do not discuss the possible alternative scenarios but make a series of claim without any real factual support. Moreover, the use of the calculations to support the interpretation of the various peaks is far from being convincing. This is a pity given that certain measurements appear as interesting.

Response: We thank the reviewer for pointing out the interesting aspects of our measurement. Concerning the interpretation of various peaks, we did not propose new assignments. Instead, we used the well accepted assignments of those peaks in the literature: 220 cm^{-1} (Nat. Commun. 2017, 8, 14670; Phys. Rev. B 2015, 91, 19541), 395 cm^{-1} (2D Mater. 2015, 2, 035003), and 406 cm^{-1} (Phys. Rev. B 2012, 85, 161403; Chem. Soc. Rev. 2015, 44, 2757). However, in this work, we found the interesting variation of these peaks in defects and used these peaks to reveal the unique electronic properties of defects for the first time. Following the reviewer's suggestion and during the past several months, we performed extensive calculations shown below and given in Section 3 of supplementary information, which would well explain our experimental phenomenon and support our interpretation.

We performed the following calculations:

- (1) We systematically calculated the band structures and Fermi levels of 1L and 2L armchair nanoribbon (ANR) with a series of width (up to 6.2 nm).
- (2) We calculated the electrostatic potential of ANR as a function of the ribbon width and obtained their work functions (i.e. Fermi level referred to the vacuum level). From the width dependent work function, we can find that there is charge transfer between the edge and basal plane, in consistent with enhancement of PL and A_{1g} mode at the edge.
- (3) We calculated the charge density of the related electronic states as a function of the ribbon width to visualize the edge and bulk states and further identify c_Q and c_K states in 1L and 2L ANR. The result well interprets the absence of LA(M)+TA(M) mode at the 1L edge and large enhancement at the 2L edge.
- (4) To confirm the different effect of the armchair and zigzag edge on the frequency shift of A_{1g} mode, we calculated the phonon dispersion and off-resonant Raman spectra of ANR and zigzag nanoribbon (ZNR) with different widths. The results agree with the experimental phenomenon.

Certain points should be addressed before any kind of publication:

- 1) Concerning the band at 220 cm^{-1} . The authors attribute it to a double resonant mechanism. One of the arguments is that it is not observed in the wrinkle defect (Fig. 1) because "the wrinkle ... does not fulfill the momentum conservation". This is bad English. I guess that the authors want to say that the wrinkle-defect does not break the translational invariance symmetry which allows to relax the momentum conservation allowing double resonant processes to happen (as in the case of

the graphite/graphene D defect). The problem is that this is not the case: a wrinkle is expected to break translational invariance symmetry. The fact that the 220 peak is observed in the presence of dangling bonds and not for the wrinkle is an indication that this peak is possibly associated with a localized vibration?

Response: We are sorry for the unclear expression. The Raman peak at $\sim 220\text{ cm}^{-1}$ have been assigned as the LA mode in previous reports (Nat. Commun. 2017, 8, 14670; Phys. Rev. B 2015, 91, 195411), which can be observed in the defective MoS₂ but is absent in the pristine MoS₂. This mode is similar to the defect-induced D mode in graphene, which comes from a DRRS process involving one phonon, where momentum conservation is provided by the elastic scattering of the excited electron by a defect. In the wrinkled graphene, the D mode appears with the lattice defect associated with the local delamination or high curvature to locally enable Raman process of D mode (Nano Lett. 2015, 15, 5098–5104), but disappears in the small curvature wrinkle (Nano Lett. 2012, 12, 3431). We were using AFM contact mode in the TERS experiment, MoS₂ wrinkle was identified as a small curvature. According to the frequency shift of the strain sensitive E_{2g} mode (Supplementary Fig. S7), the strain of the wrinkle is small ($\leq 0.4\%$), which exhibits little influence on the lattice vibration (Nat. Commun. 2017, 8, 1370) and electron band structure (Fig. 2b). Thus, this wrinkle induced defect assisted elastic scattering can be neglected. Consequently, the LA mode is absent in wrinkle with a small curvature. For clarity, we revised the mentioned sentence “*However, the wrinkle will only result in lattice strain rather than atoms with dangling bonds and does not fulfill the momentum conservation.*” in the main text as follows:

However, the investigated wrinkle induces a small lattice strain ($\leq 0.4\%$) (Supplementary Fig. S7), which exhibits little influence on the lattice vibration. Consequently, the wrinkle assisted elastic scattering can be neglected, and the LA mode in wrinkle with a small curvature is absent according to the Raman selection rule.

2) Fig. 2b reports the electronic bands calculated for a nanoribbon of finite width. In this kind of plot (well understood and commonly used in literature) one observes an important number of bands that correspond to "bulk" electronic bands + certain bands associated to states localized at the edge. For a finite-width ribbon, the "bulk" bands are affected by quantum confinement effects and, for example, the apparent electronic gap will be different from that of the bulk and will change in a visible way by increasing the ribbon width. The relative position of the conduction and valence bands in Figure 2 is thus strongly dependent on the width of the ribbon. This kind of study makes sense only if it is done on several different ribbons with varying width. This point is relevant since the bands of Figure 2 are used in the interpretation of the measurements.

Response: We thank the reviewer for the kind suggestion. We followed the reviewer' suggestion and calculated electronic properties that are used in this study for nanoribbons with a series of widths (10~30 Å). In this way, we can obtain the edge effect on the “bulk” states involved in DRRS as a function of distance from the edge. We obtained their band structures and determined the relative position of c_K and c_Q states as a function of the ribbon width by the fat-band analysis. From the fitting results as presented in Fig. R4, E_{CK}-E_{CQ} converges to a constant value with the increase of the ANR width to larger than 28 and 32 Å in 1L and 2L ANR, respectively. The possibility of DRRS process of LA(M)+TA(M) mode at the edge of 2L MoS₂ can be largely

enhanced compared with that of 1L edge because of the larger $E_{CK}-E_{CQ}$, leading to the observation of the LA(M)+TA(M) mode at the edge of 2L MoS₂. For clarity, we added all the calculation details and discussion into the revised Supplementary Section 3.

Figure. R4 The energy difference between c_K and c_Q states ($E_{CK}-E_{CQ}$) in 1L and 2L ANR as a function of the nanoribbon width.

3) If the authors want to claim that the band gap is affected by electronic states at the edges they should prove the present of surface electronic states in at the minimum/maximum of the conduction/valence bands. This cannot be evinced from Figure 2 (actually it looks quite unlikely from that figure).

Response: Fig. 2 was expected to determine the edge effect on the relative position of DRR involved electronic states in the conduction band, which is related to the enhancement of LA(M)+TA(M) mode at the edge, as discussed in Section 3 of the revised supplementary information. The bands existing in the gap of the bulk bands result from edges, which would exhibit little contribution to the DRR process. Following the reviewer's suggestion, we further calculated the band structure of 1L ANR with a large width up to ~ 62 Å ($n=39$, 1L 39ANR) and pristine MoS₂ with the fat-band analysis as well as the charge density of related states, as shown in Fig. R5. The charge densities of the minimum of the conduction band (c_{K1}) and the maximum of the valence band (v_K) are localized near the edge, indicating that the surface electronic state is at the minimum of the conduction band and maximum of the valence band and band gap of ANR is determined by surface electron states. Furthermore, the charge densities of $c_{K(1-6)}$ states also indicate that bands in the gap of bulk bands originate from the edge. The charge densities of $c_{K(7-10)}$ and $c_{Q(7-10)}$ states are localized, which indicates the degeneration of c_K states and reveals the edge effect on the bulk states. To confirm the edge effect on the electronic states involved in DRR process, we further calculated the band structure of 1L and 2L ANR with varying width (10-30 Å), and $E_{CK}-E_{CQ}$ as a function of distance from the edge can be derived, as discussed in the 2nd question of Reviewer 2. For clarity, we added all the calculation detail and discussion into the revised Supplementary Information as section 3.2.

Figure R5 (a) Schematic representation of MoS₂ nanoribbon with the armchair edge. Band structure for 1L MoS₂ (b) and 1L 39ANR (c), where the Fermi level is set at zero. The fat bands with the projected weight of Mo atomic d_{z^2} and $d_{x^2-y^2}$ orbitals within the eigenstates. Charge density isosurfaces of electron states of 1L MoS₂ (d) and 1L 39ANR (e), specified in b and c, respectively.

4) Figure 2. In panel b there are four electronic bands. It is relatively clear how the bands labeled "1L" and "2L" have been obtained. It is not clear how "wrinkle" and "1L-2L step" calculations were done. I could not find an understandable description neither in the text nor in the Supplemental Information.

Response: We are sorry for not providing calculation details about wrinkle and 1L-2L step. Here, the wrinkle in MoS₂ is simplified into 1L MoS₂ with 0.4% uniaxial strain along b axis, according to the frequency shift of E_{2g}^1 mode (Supplementary Fig. S7). The 1L-2L step can be constructed by a vertical combination of 1L edge and 1L basal plane, where 1L edge can be simplified as a nanoribbon. Consequently, the 1L-2L step is modeled by a combination of 1L ANR and 1L basal plane. To avoid interaction between periodic ANR, lattice spacing along b axis is larger than 10 \AA , as shown in Fig. R6b. The corresponding illustrations have been added in the revised Supplementary Information as section 3.1.

Figure R6 Schematic models of the wrinkle (a) and 1L-2L step (b) in MoS₂.

5) Line 178/179. "The Fermi level at the edge is higher than that of the basal plane, leading to charge transfer...". It is not clear to me how the change in Fermi level observed in the calculations (done in vacuum) could possibly be compared to a depletion possibly present in the measurements (done on flakes lying on a surface). How do the work functions of the structures of Fig. S5b? How are they affected by the presence of a substrate?

Response: In fact, there is no charge transfer between MoS₂ and substrate, as discussed in our response to the first question of Reviewer 1 due to the existence of the unavoidable thin ice-like water layer under the constraint condition even at room temperature. The edge bands result from dangling bonds with the translational symmetry along the one-dimensional edge. The occupied edge bands overlap with valence bands of the basal plane, and band alignment between them results in charge transfer between the edge and basal plane. This charge transfer leads to the change of the electron density near the edge, which changes the intensity of A_{1g} mode. To investigate the effect of edge on the charge transfer, it is necessary to have the work functions of the edge and basal plane. We first calculated the electrostatic potential of MoS₂ and ANR with a series of width, and 1L 20ANR is taken as an example to illustrate the approach to obtain the vacuum level (E_{vac}), as shown in Fig. R7a. The work function (E_W) can be obtained by $E_W = E_{vac} - E_F$, where E_F is the Fermi level obtained from the band structure calculation. E_W as a function of ribbon widths is presented in Fig. R7b, where E_W of ANR is obviously larger than that of the basal plane, indicating the charge transfer from the edge to the basal plane. This charge transfer results in depletion of electrons near the edge, which lead to the increase of the intensity of A_{1g} mode. The corresponding discussion has been added in Section 3 of the revised Supplementary Information.

Figure R7 (a) The electrostatic potential energy of 1L 20ANR along the z axis. (b) The Fermi

level referred to the vacuum level, i.e., the work function, of 1L and 2L ANR as a function the ribbon width.

6) Figure 4, panel c. Concerning the calculations. How are these numbers calculated? The authors simply provide a couple of points in the figure without describing how they were obtained. The authors simulated some ribbons and then they calculate the phonons. The phonons that you can obtain with a ribbon calculations have the same problems of interpretation of the electronic bands discussed above. Are we talking about vibrations localized at the edge? How can you decouple the two edges of the ribbon by using ribbons with such a small width? Are we talking about vibrations of the bulk? How are these vibrations affected by the confinement of the vibration within the width of the ribbon? In their structures the authors have tens of vibrational modes with overlapping frequencies. What is the criterion chosen to plot the points reported in Figure 4C? This kind of study makes sense only if ribbons with different widths are studied as a function of the width. Using just one ribbon can be misleading.

Response 6: There are several questions in this comment. Therefore, to make it clear, we divide them into 4 sub-questions, and answer them point by point.

Question 6-1: Figure 4, panel c. Concerning the calculations. How are these numbers calculated? The authors simply provide a couple of points in the figure without describing how they were obtained.

Response 6-1: This question is concerned with how we calculated the frequency of A_{1g} mode for different MoS_2 structures. In previous manuscript, we calculated the phonon dispersions and vibration density of states (VDOS) by Phonopy software (Scr. Mater. 2015, 108, 1-5) for different types of MoS_2 . It is challenging to reliably assign calculated modes only on the basis of the obtained frequency, because there are tens of vibrational modes showing similar frequencies to that of A_{1g} mode. In the revised version, we employed more rigorous density functional perturbation theory (DFPT) using VASP code to calculate phonon frequency at Γ point of different types of MoS_2 and Raman intensity ($\alpha_{xx}^2 + \alpha_{xy}^2 + \alpha_{yx}^2 + \alpha_{yy}^2$) with Raman off-resonant activity calculator (A. Fonari and S. Stauffer, vasp_raman.py, <https://github.com/raman-sc/VASP/>, retrieved 2013). We then calculated the pristine MoS_2 (in supercells repeated along the armchair direction), ZNR (with width of 25.1 Å, 30.5 Å, and 35.9 Å), and ANR (with width of 15.6 Å, 17.4 Å, and 20.5 Å). A similar procedure has been employed to successfully reproduce experimental Raman spectra of MoS_2 with and without defect and further estimate the influence of defect on the lattice vibration [Phys. Rev. Appl. 2017, 7, 024001]. The detailed results will be shown in our response of Question 6-2. It can found later that the current method gives the same conclusion as that in the previous manuscript.

Question 6-2: The authors simulated some ribbons and then they calculate the phonons. The phonons that you can obtain with a ribbon calculations have the same problems of interpretation of the electronic bands discussed above. Are we talking about vibrations localized at the edge? How can you decouple the two edges of the ribbon by using ribbons with such a small width? Are we talking about vibrations of the bulk? How are these vibrations affected by the confinement of the vibration within the width of the ribbon?

Response 6-2: This question is concerned with whether the width is large enough for a reliable calculation of phonons. As stated in our response to Question 1 that we have performed a systematical calculation of the electronic and phonon properties of the nanoribbons with varying width until the results become convergent. Different from the electronic properties, the vibrations are localized at the edge and can be visualized in real space via the atomic displacement of the related modes. Calculation of vibrational properties is more resource-demanding, and it is too expensive and time-consuming to perform the width-dependent calculation for both ANR and ZNR. Instead, following the suggestion of the reviewer, we added in section 3.4 of the revised Supplementary Information the discussion about the phonon at Γ point and corresponding lattice vibrations of ANR with widths of 15.6, 17.4, and 20.5 Å and ZNR with widths of 25.1, 30.5, and 35.9 Å, and further calculated their off-resonance Raman intensity, as discussed in Response 6-1. As will be found below, for the pristine MoS₂, the A_{1g} mode is a vibration of the bulk. At the edge of MoS₂, this vibrational mode is a localized mode. This mode is almost independent of the width of nanoribbon in both ANR and ZNR, indicating the width we used is large enough to avoid the coupling effect between two edges of the nanoribbon. The new calculation also shows the downshift and upshift for the A_{1g} mode at zigzag and armchair edges, respectively, confirming the conclusion in the previous manuscript.

To make our point clear, we present the calculation details below as well as in section 3.4 of the revised Supplementary Information:

The mode with a similar atomic displacement to the A_{1g} mode of the nanoribbon is denoted as A_{1g}-like mode. The mode with vibration localized at the edge is indicated in blue and the ones on the basal plane near the edge are indicated in orange (Fig. R8).

For ZNR, there are S and Mo edge. Raman intensities near the two edge are not comparable, and their Raman spectra are presented separately in Fig. R8b. According to calculated Raman intensities, there are three A_{1g}-like modes, with vibration localized at S edge (~399 cm⁻¹), Mo edge (~402 cm⁻¹), and with vibration component on basal plane near edge (~406.5 cm⁻¹). The frequencies of all these three A_{1g}-like modes exhibit negligible difference at the different nanoribbon width, indicating the decoupling between two edges. The higher frequency modes at ~407 cm⁻¹ in ZNR show a low intensity, and the mode might result from band folding and splitting. It is obvious that the zigzag edge (no matter Mo or S edge) has a lower frequency than that of the basal plane, agreeing with the downshift of A_{1g} mode in experiment.

For ANR, there are two A_{1g}-like vibration modes localized related to the edge and basal plane. The former has a higher frequency (~413 cm⁻¹) than the latter (~406 cm⁻¹), as presented in Fig. R8c.

Figure R8 Calculated Raman spectra and lattice vibration of 1L pristine MoS₂ (a), ZNR (b) and ANR (c) with different widths, where vibration modes are obtained with VASP and Raman intensities are obtained with the Raman off-resonant activity calculator using VASP as a back end. Envelopes of each Raman spectra are evaluated by smearing the peaks with a half width of 5 cm⁻¹. Note that the Raman intensities near the Mo and S edge are not comparable.

For clarity, we modified Figure 5 and the related discussion in the “Determination of different edge structures” part of the revised manuscript as follows:

“The calculation result given in Fig. 5c shows that the frequency of the A_{1g} mode at the armchair edge would upshift compared with that on the basal plane.”

Figure 5 | Effect of the edge structure on the peak position of the Raman A_{1g} mode. (a) Typical line-trace TERS spectra of zigzag edge (left panel) and armchair edge (right panel) in the spectral range of the A_{1g} mode. Note that the spectra have been normalized with the intensity of A_{1g} peak for a better comparison. (b) Plots of peak position with the tip position. The corresponding line-trace TERS spectra are shown in Supplementary Fig. S16a and Fig. S20a, respectively. The dash red lines are guides for the eye. (c) Calculated Raman spectra and lattice vibration of the basal plane, zigzag nanoribbon (ZNR, with a width of 35.9 Å) localized at the Mo and S edges, and armchair nanoribbon (ANR, with a width of 20.5 Å). The envelope of each Raman spectrum is evaluated by smearing the peaks with half width 5 cm⁻¹. Note that the displacements of A_{1g} or A_{1g}-like mode with out-of-plane vibration are side viewed. See Supplementary Section 3.4 for details.

Question 6-3: In their structures the authors have tens of vibrational modes with overlapping frequencies. What is the criterion chosen to plot the points reported in Figure 4C?

Response 6-3: As discussed in Response 6-1, the criterion chosen to plot points is the lattice vibration of phonon at Γ point, which is similar to that of A_{1g} mode in the basal plane. In the revised manuscript, we further calculated Raman intensity of the related mode in the basal plane and nanoribbons. The frequency difference between A_{1g} mode in the basal plane and nanoribbons indicates frequency shift of A_{1g} mode at edge, with a lower frequency at the ZNR edge and the higher frequency at the ANR edge compared with that in basal plane.

In section 3.4 of revised supplementary information, we discuss the frequency shift of A_{1g} mode according to the calculated Raman spectra, which provides convincing and intuiting evidence for the edge effect on A_{1g} mode.

Question 6-4: This kind of study makes sense only if ribbons with different widths are studied as a function of the width. Using just one ribbon can be misleading.

Response 6-4: We understand that it would be ideal if the width dependent vibrational information can be calculated. However, calculation of the phonon dispersion and Raman intensity of ANR and ZNR for a certain width (~ 40 atoms and large vacuum space along a/b (>10 Å) and c (>20 Å) direction) already takes weeks on a massively parallel system with tens compute nodes (24 cores per compute node). Therefore, it is too demanding to calculate the vibration properties of ANR and ZNR with a series of width to consider the edge effect due to the limited computational resource. Alternatively, we calculated just two widths as discussed in Response 6-1 and 6-2, which still gave the same trends, i.e. A_{1g} mode at zigzag and armchair edges exhibit downshift and upshift, respectively.

7) Calculations on the ribbons. The details of the calculations are not enough. What is the lattice spacing? The structures of the ribbons are described in Figure S10. What is the criterium to distinguish among "reconstructed bonds" and "pristine bonds"? If you repeat the same calculation (geometry optimization) with a wider ribbon, how are the bond lengths affected? (this kind of calculations is not particularly heavy, but it is part of the standard tests done with these kind of problems). The materials reported in the manuscript is not enough to judge about the reliability of these calculations.

Response: Thanks for the meaningful comments and useful suggestion. The calculation details have been added in section 3 of revised Supplementary Information. The lattice spacing between periodic nanoribbons along a/b and c directions are larger than 10 and 20 Å, respectively, to avoid interaction between periodic nanoribbons. In the previous manuscript, the criterion to distinguish among "reconstructed bonds" and "pristine bonds" is the length of the bond. The length of pristine bonds can be obtained from the relaxed structure of pristine MoS_2 , and the length of reconstructed bonds obtained from the relaxed structure of nanoribbon is significantly different from that of pristine bonds. However, it is insufficient to investigate edge effect on the intensity of LA(M)+TA(M) mode, because it is difficult to create a relation of the lattice disorder with the intensity of LA(M)+TA(M) mode. As discussed in our response to the second question of Reviewer 2, the intensity variation of LA(M)+TA(M) mode at the edge is attributed to the influence of edge induced band bending on DRRS involved states (i.e. K and Q states in

conduction band) (Fig. R4). Thus, we modified the discussion about the intensity profile of LA(M)+TA(M) mode in the revised manuscript on page 9 line 6 as follows:

“Thus, the intensity variation region of LA(M)+TA(M) mode at the edge can reflect the band bending region of DRRS involved states (i.e. K and Q states in conduction band) (see Supplementary Section 3.3 for more details). We plot its intensity as a function of tip position as the red curve in Fig. 3c. The intensity becomes higher when the tip was moved closer to the edge, indicating a stronger DRRS involved band bending at the edge. The full width at half maximum (FWHM) of the 396 cm^{-1} intensity profile is 7 nm (Supplementary Fig. S5, and see Supplementary Section 2 for more details). It should be pointed out that the measured TERS intensity profile is a convolution of the DRRS involved band bending region ($d_{\text{LA(M)+TA(M)}}$) and the spatial resolution of TERS. Considering the typical spatial resolution of AFM-based TERS obtained in our lab of about 7 nm, which is equal to the experimentally measured TERS intensity profile, we expect the length of the DRRS involved band bending region should be much smaller than 7 nm. Indeed, theoretical calculation reveals that $d_{\text{LA(M)+TA(M)}}$ of bilayer MoS₂ induced by the edge is only about a few lattice spacing of 1.8 nm (shown in Supplementary Fig. S12).”

Minor points:

8) Figure2. In the first lines of the caption the authors talk about strain and then refer to Figure S6 of the Supplementary Info for more details. Figure S6 does not seem to be related to this discussion.

Response: Thanks for pointing out the problem and we have revised the caption of Figure S6.

9) Throughout the paper the authors use the word "reconstruction" to refer (I guess) to atomic relaxation present near the edges of the structure. In surface science, the term "reconstruction" has a specific meaning which does not correspond to this. To avoid confusion they should avoid the use of the term "reconstruction".

Response: Thank you for pointing out the concepts of reconstruction and relaxation. We carefully look into the classic surface science book. In surface science, surface relaxation means that a bulk crystal periodic arrangement remains the same at the surface as it is in the bulk. But the first interlayer spacing is modified (Fig. R9a) or the first layer parallel displaces relative to the second layer (Fig. R9b). Surface reconstruction means that the surface atomic arrangement changes and shows a different symmetry and periodicity from that of the bulk lattice cell underneath (Fig. R10). (Oura, K. et al. Surface Science: An Introduction, 171-194 (Springer Berlin Heidelberg, Berlin, Heidelberg, 2003). In our work, edges lose atoms and form dangling bonds, the atomic arrangement significantly change. Therefore, in our case, it is indeed reconstruction rather than relaxation. The term of “reconstruction” has been widely used in describing the defect structure of 2D materials. (Nano Lett. 2018, 18, 482–490; J. Am. Chem. Soc., 2012, 134, 6204–6209; ACS Nano, 2013, 7, 4495–4502). Therefore, we think it is better to retain the term of reconstruction for the changed lattice structured at the edge.

Figure R9 Schematic illustration of (a) normal and (b) lateral relaxation in the top atomic layers of a semi-infinite crystal. Copied from Oura, K. et al. *Surface Science: An Introduction*, 171-194 (Springer Berlin Heidelberg, Berlin, Heidelberg, 2003).

Figure R10 Schematic illustration of (a) conservative and (b) non-conservative reconstruction in the top atomic layers of a semi-infinite crystal. Copied from Oura, K. et al. *Surface Science: An Introduction*, 171-194 (Springer Berlin Heidelberg, Berlin, Heidelberg, 2003).

Reviewer #3 (Remarks to the Author):

In this work, the authors demonstrate the capabilities of TERS to identify spectral features that exist only near defects. The experimental work seems to have been carried out with care, and the data should attract some interest from researchers in the related fields. However, there are a few issues that prevents this reviewer from recommending publication of this work in *Nature Communications*.

First of all, the importance or impact of the work is questionable. It turned out that the main focus of this work is the edge of 2-layer MoS₂. The authors explain that the special band reconstruction near edges of 2L MoS₂ allowed the defect-related Raman features to show up. This means that this method is useless for monolayer MoS₂ which is far more interesting and important or for other thicknesses. In this sense this work would have a very limited impact which is very disappointing.

Response: In our previous version, we used our method to reveal the interesting phenomenon at the edge of bilayer MoS₂, which may find important application in energy conversion and storage devices and catalysis (*Nano Lett.* 2012, 12, 4674; *Nat. Nanotechnol.* 2016, 11, 421; *ACS Nano* 2018, 12, 1592). But this does not mean that TERS cannot be applied to the monolayer sample. In fact, we have already applied our method to investigate the electronic transition regions induced by defects in monolayer MoS₂ (monolayer edge, step edge between monolayer and bilayer), as

shown in Fig. R11. We found a much larger electronic transition region (~18 nm) induced by the monolayer edge (Fig. R11c) than that induced by bilayer edge (~10 nm, Fig. 3c inset). Interestingly, we observed two obviously different electronic transition regions of (~16 nm) and (~19 nm) to the side of 1L and 2L MoS₂ respectively, near the step edge (Fig. R11d). These studies clearly revealed the defect-induced electronic transition region of MoS₂ in the real application environment, which may provide a guide in engineering the defect density to achieve desired properties. We think this result may also be interesting to the readers and thus add the Figure R11 as Figure 4 and the following discussion into the main text at Page 10, line 11:

“We further used TERS to investigate electronic transition regions induced by the monolayer edge and the step between monolayer and bilayer (1L-2L) MoS₂. As shown in Fig. 4a and b, no 396 cm⁻¹ signal can be detected in the monolayer edge, and the 396 cm⁻¹ intensity fluctuates in a narrow region (FWHM~6.9 nm) across the 1L-2L step indicates a high TERS spatial resolution in these measurements. Similar to the bilayer edge, the electron density becomes smaller when the tip was moved closer to the edge and the step. The lowest electron density (i.e. the maximum A_{1g} intensity) appears inside the pristine MoS₂ and is located in a few nanometers away from the edge (Fig. 4c) and the step (Fig. 4d). In addition, we can obtain a larger electronic transition region of ~18 nm induced by the monolayer edge (Fig. 4c) than that of 10 nm induced by the bilayer edge (Fig. 3c inset). This may be a result of a larger Fermi energy difference between the monolayer edge and the basal plane (Supplementary Fig. S13b) as well as a deeper oxygen p-type doping due to the higher chemical activity compared with that of 2L MoS₂⁴¹. Interestingly, we observed two obviously different electronic transition regions near the 1L-2L step (Fig. 4d). The step is doped by oxygen and acts as a narrow quantum wire⁴⁰, and thus induces electronic transition regions near the step in both monolayer (~16 nm) and bilayer (~19 nm). The above study clearly reveals the defect-induced electronic transition region of MoS₂ in the real application environment, which guides better engineering of the defect density to achieve desired properties.”

Figure R11 (Figure 4) Electronic transition regions induced by 1L edge and 1L-2L step. Plots of normalized intensities of TERS peaks with the tip positions across the 1L edge (a) and 1L-2L step (b). The solid lines are the fitted results. See Supplementary Section 2 for details about the data fitting. Note that these spectra are the pure near field signals and have been subtracted with the far field signals. Supplementary Fig. S20b and c show more data sets for the corresponding line-trace TERS spectra. Enhanced TERS intensity profiles of the electronic transition region in 1L edge (c) and 1L-2L step (d) after the deconvolution of the EM field intensity distribution (see Supplementary Section 2 for more details).

Then there are issues of interpretation. Some of the interpretations are not based on experimental evidence but on inference based on calculations. Since there are experimental means to verify the claims, the authors should strive to back up their claims with experimental evidence.

1) In page 10, the authors claim that the maximum intensity position of the A_{1g} mode is located a few nanometers away from the edge. However, the two vertical lines in Fig. 3c are separated by less than 1 nm, much smaller than the spatial resolution of TERS. This small difference might be caused by the asymmetric profile of the A_{1g} intensity (blue curve).

Response: We are sorry that we did not make this point clear. Fig. 3c is the original data of A_{1g} intensity with the tip position without the deconvolution of EM field. After the deconvolution, we can find that the maximum intensity of A_{1g} mode (the lowest electron density) appears inside the pristine MoS_2 and is located at 4 nm away from the edge (Supplementary Fig. S4b). In a separate experiment, we clearly observed such a separation of the two maximum intensity positions from the original data (Fig. R12a). Furthermore, in our electrochemical TERS measurement of the system, we found that this separation becomes obvious at the hydrogen evolution potential

(unpublished work). Therefore, we think the separation is indeed an intrinsic property of MoS₂ edge and thus carefully placed the two vertical lines at the maximum in Fig.3c. For clarity, we used the Supplementary Fig. S4b as an inset of Fig. 3c and add Fig. R12 into the Supplementary Information and rephrase the related part on page 10 line 5 and also as follows:

“the maximum intensity of A_{1g} mode (the lowest electron density) appears inside the pristine MoS₂ and is located at just a few nanometers away from the edge (Fig. 3c inset, Supplementary Fig. S19).”

Figure R12 (a) Plots of normalized intensities of two TERS peaks (396 and 406 cm⁻¹) of bilayer MoS₂ with the tip position. The solid lines are the fitted results. Note that these spectra are the pure near field signals and have been subtracted with the far field signals. (b) Enhanced TERS intensity profiles of the electronic transition region induced by bilayer edge after the deconvolution of the EM field intensity distribution.

2) The analysis of the zigzag vs armchair edges is not fully substantiated by experimental data. It seems that the identification of the edge type is based on DFT calculations only. The authors should find ways to back up their claims with experimental data such as TEM, STEM, STM, SHG, etc. There are many ways to determine the edge types (approximately). Another way is to compare the results with samples grown by CVD. In CVD samples, the edge types are often determined from the shape of the platelets.

Response: Thanks for the kind suggestion. It will be ideal to verify the atomic arrangement by an imaging method. However, it is a great challenge for us to back up our claims by TEM, STEM, or UHV-LT STM. The mechanical exfoliated edge is not a pure zigzag or armchair edge but that mixed with each other in nanometer scale (Niimi, Y. et al. Phys. Rev. B 2006, 73, 085421; Tinoco, M. et al. Nano Lett. 2017, 17, 7021-7026). Thus, it is difficult to locate the same position of the edge in TERS and STEM/STM experiments as well as to correlate of these two experimental results. Alternatively, in this work, we carefully took all possible factors into consideration before making the claim and also used DFT calculation to further back up the conclusion. We hope that this conclusion can be conveniently used to identify the edge type of TMDC materials under the ambient condition. Recently, we are looking for some good CVD samples with well-defined edge types and will try to correlate the edge type with the A_{1g} position as well as to investigate the evolution of TERS spectra of different edges during the electrocatalytic process (e.g. hydrogen evolution reaction).

Also, some of the explanations are not correct or incomplete.

1) On page 4, the authors claim, “the wrinkle will only result in lattice strain rather than atoms with dangling bonds and does not fulfill the momentum conservation.” Any defect that destroys the translational symmetry of the lattice would result in relaxation of the momentum conservation. There is no difference in the dangling bonds and local strain in this regard. The authors should find some other plausible explanation to explain the absence of the LA feature at wrinkles.

Response: As discussed in question 1 of reviewer 2, the Raman peak at $\sim 220\text{ cm}^{-1}$ has been assigned as LA mode in previous reports (Nat. Commun. 2017, 8, 14670; Phys. Rev. B 2015, 91, 195411), which can be observed in the defective MoS₂ but is absent in the pristine MoS₂. This mode is similar to the defect-induced D mode in graphene, which comes from a DRRS process involving one phonon, where momentum conservation is provided by the elastic scattering of the excited electron by a defect. In the wrinkled graphene, the D mode appears with the lattice defect associated with a local delamination or high curvature to locally enable Raman process of D mode (Nano Lett. 2015, 15, 5098–5104), but disappears in the small curvature wrinkle (Nano Lett. 2012, 12, 3431). We were using AFM contact mode in the TERS experiment, MoS₂ wrinkle was identified as a small curvature. According to the frequency shift of the strain sensitive E_{2g} mode (Supplementary Fig. S7), the strain of the wrinkle is small ($\leq 0.4\%$), which exhibits little influence on the lattice vibration (Nat. Commun. 2017, 8, 1370) and electron band structure (Fig. 2b). Thus, this wrinkle induced defect assisted elastic scattering can be neglected. Consequently, the LA mode is absent in wrinkle with a small curvature.

2) The explanation for the LA(M)+TA(M) mode sounds plausible. Then, why the LA(M)+LA(M), in other words, the 2LA(M) mode is not enhanced at the edges? One would think that the latter is more likely excitation than the LA+TA scattering.

Response: The 2LA(M) mode has been assigned by Carvalho et al. (Ref. 21, Carvalho, et al. Nat. Commun. 2017, 8, 14670) and is located at $\sim 470\text{ cm}^{-1}$, which merges with the other two modes (p_2 and p_3) to form a broad Raman peak (Fig. R13). By deconvolution, we did see the intensity of 2LA(M) enhanced by about 2 times at the edge due to the DRRS. On the other hand, as discussed in the previous manuscript, the 396 cm^{-1} Raman peak has been observed in the bulk MoS₂ at an ultralow temperature (Phys. Rev. B 2010, 81, 195209) and assigned as LA(M)+TA(M) (2D Mater. 2015, 2, 035003). In this manuscript, we observed this mode in the defect for the first time and found that this mode related to a DRRS process.

Figure R13 Two typical TERS spectra of a bilayer MoS₂ at the basal plane and the edge.

3) On the bottom of page 5, the authors explain the reconstruction of the electronic band near the edge defects. However, the band picture is not strictly valid in the presence of defects and the momentum conservation would be significantly relaxed, especially when the reconstruction is limited to the region of only a few nanometers in width.

Response: As the reviewer comments, edge effect determined by an ANR with finite width is not strictly valid. We carefully re-considered the theoretical analysis, and added calculation to obtain band structure of nanoribbons with a series of width (1~3 nm). We then obtained the edge effect on the band reconstruction, where c_K and c_Q states involved in DRRS in band structure of ANR are determined by fat-band analysis and charge density of related electron states, as presented in section 3 of supplementary information. Accordingly, we obtained $E_{c_K} - E_{c_Q}$ as a function of distance from the edge, as shown in Fig. R14, which can characterize the possibility of DRRS process. It is found that the relative energy of c_K and c_Q states are strongly dependent on the width and layer number of nanoribbons, which indicates the edge effect on the intensity of LA(M)+TA(M) mode. The calculations on a series of ANR with fat-band analysis and charge density of related electron states provide strictly valid edge effect on the band structure, where the obtained relative energy of c_K and c_Q states as a function of distance from the edge explains intensity variation of LA(M)+TA(M) mode near the edge.

As the reviewer mentioned, the region of only a few nanometers in width would relax momentum conservation and further result in phonon confinement effect. This work focuses on the intensity of LA(M)+TA(M) mode, which is activated by DRRS process and can also be observed in pristine 3L and 4L MoS₂, as presented in Fig. S15d. As previously reported (Phys. Rev. B 2015, 91, 195411) relaxation of momentum conservation and phonon confinement effect decrease the intensity of the Raman mode, which is in contrast to enhancement of LA(M)+TA(M) mode at the edge, indicating little influence on the intensity of LA(M)+TA(M) mode.

Consequently, the intensity enhancement of LA(M)+TA(M) mode at edge is attributed to edge induced band bending, which is obtained by band structure evolution of ANR with varying width, and the influence of momentum conservation relaxation is neglected.

To clarify that, the discussion about relaxation of momentum conservation and phonon confinement effect on the intensity of LA(M)+TA(M) mode has been added in section 3 of supplementary information.

Figure R14 The energy difference between c_K and c_Q states ($E_{CK}-E_{CQ}$) in 1L and 2L ANR as a function of the nanoribbon width.

4) I do not understand the following phase in page 9: ‘... depletion region between the metallic edge and the semiconducting basal plane.’ The edges are more intrinsic than the basal plane because the p-doping depletes the excess electrons that are present in the basal plane. What is the meaning of ‘metallic edge’? How can one see an exciton in a metallic material?

Response: The zigzag and armchair edges of MoS₂ are metallic and semiconducting, respectively, as described in Ref. 9. We agree with the reviewer that excitons do not exist in metallic materials. However, edge states are localized at the edge and only confined to just a few lattices, as described in our response to Question 3 of Reviewer 2. However, away from the edge, the bulk states dominate its properties, where electron charge density is lower than that in the base plane of MoS₂. Consequently, the exciton can be observed near the edge.

To clarify it, we have modified expression in the revised manuscript as shown in the Question 1-3 of Reviewer 1.

Some additional comments:

1) It would be nice to include TERS (not far field) spectra pristine 1L and 2L basal planes in Fig. 1c.

Response: We thank the reviewer for the kind suggestion and have added the near field spectra of pristine 1L and 2L basal planes into the Fig. 1c (upper two spectra), as shown below.

Figure R15 Raman spectra of four 1D defects and the basal plane in MoS₂ marked in (b) when the tip was approached and retracted. These spectra are the near field spectra and have been subtracted

with the far field signal as well as the tip-enhanced photoluminescence background from MoS₂ and SPR from the TERS tip. The intensity is normalized to the A_{1g} peak for comparison.

2) It is well known that there are very high densities of S vacancies even in exfoliated MoS₂. It would be nice if the authors can isolate single S-vacancy with TERS and see if any spectral feature can be correlated.

Response: We thank the reviewer for the kind suggestion. Indeed, S vacancies have received more and more interests these years. However, it is still difficult to characterize the single S vacancy even by TERS in the ambient environment. First, a single S vacancy cannot be found by the conventional AFM imaging due to the thermal drift and the large radius of the AFM-TERS tip ($R_{\text{tip}} = 65 \text{ nm}$). Second, the spatial resolution of AFM-TERS in air (7 nm) is much larger than a single S vacancy, which may not provide a high contrast of near field/far field signals. We agree that this is an important and interesting problem for us to collaborate with the TERS groups with an ultra-high vacuum and ultra-low temperature to visualize the correlated spectral feature of single S vacancy.

Reviewers' Comments:

Reviewer #1:

Remarks to the Author:

I would like to thank the authors for their responses to my questions and for clarifying the unclear points. I believe that the revised version of the paper is suitable for publication. The work is novel and of interest for the scientific community.

Reviewer #2:

Remarks to the Author:

Dear Editor,

my main concerns about this paper were about the computational part.

The authors have made many efforts to improve this part and they have added an important amount of new data which support the conclusions based on first principles calculations. The experimental part seems an extremely relevant kind work. I have no more objections for publication.

Sincerely,

Reviewer #3:

Remarks to the Author:

The authors have revised the manuscript extensively to address the comments by the referees.

Some of the ambiguities in the previous version have been clarified, but there are a few potentially critical issues that have not been resolved.

1. It is an overstatement to claim that they 'spatially distinguished zigzag and armchair edges' from their work. The authors have not experimentally identified the edge types as this referee suggested. The authors explain that experimental identification of edge types are difficult because the exfoliated edges are often mixed types. This is true, but the authors know that the original work on the identification of edge types in graphene was done on exfoliated samples. For example, the sample in Fig. 1b has relatively straight edges. If one measures TERS and STM images on these edges, one should be able to pinpoint the correlation between the TERS signatures with the edge types. (Now I realize that the authors do not show the images of the edges they measured for Fig. 5. The AFM or optical images should be given in SI.)

2. If the claim of identification of edge types is discounted, the only remaining 'discovery' is the observation of the 396 cm^{-1} peak at the edge of bilayer MoS₂. I am not sure if it is significant enough to justify the publication of this work in this journal.

3. Minor point. The colors of the vertical dashed lines in Fig. 3c are reversed (red vs. blue). They were correct in the previous version.

Reviewers' comments:**Reviewer #1 (Remarks to the Author):**

I would like to thank the authors for their responses to my questions and for clarifying the unclear points. I believe that the revised version of the paper is suitable for publication. The work is novel and of interest for the scientific community.

Reviewer #2 (Remarks to the Author):

Dear Editor,

My main concerns about this paper were about the computational part. The authors have made many efforts to improve this part and they have added an important amount of new data which support the conclusions based on first principles calculations. The experimental part seems an extremely relevant kind work. I have no more objections for publication.

Reviewer #3 (Remarks to the Author):

The authors have revised the manuscript extensively to address the comments by the referees. Some of the ambiguities in the previous version have been clarified, but there are a few potentially critical issues that have not been resolved.

1. It is an overstatement to claim that they 'spatially distinguished zigzag and armchair edges' from their work. The authors have not experimentally identified the edge types as this referee suggested. The authors explain that experimental identification of edge types are difficult because the exfoliated edges are often mixed types. This is true, but the authors know that the original work on the identification of edge types in graphene was done on exfoliated samples. For example, the sample in Fig. 1b has relatively straight edges. If one measures TERS and STM images on these edges, one should be able to pinpoint the correlation between the TERS signatures with the edge types. (Now I realize that the authors do not show the images of the edges they measured for Fig. 5. The AFM or optical images should be given in SI.)

Response: We thank the reviewer for the kind suggestion. It would ideal if both TERS spectra and atomic resolution STM images can be obtained simultaneously. However, as stated in the previous response that in the current TERS system as well as most ambient TERS instruments, it is still very challenging to obtain an atomic resolution image of 2D materials with a plasmonic gold tip, which tends to have mobile apex atoms and significant mechanical drift. Instead of direct correlation of spectral shift with the edge image, just as what has been done in graphene case (Ref. 42) by correlating the white light image with Raman measurement, we mechanically exfoliated 1L MoS₂ to obtain a sample with defined angles of 60° and 90° between two adjacent edges (Fig. 5d). For an angle of 60° (Fig. 5e), both edges have the same structure (either zigzag or armchair). The frequency shift direction of the A_{1g} mode at both edges should be either downshift or upshift. On the other hand, for an angle of 90° (Fig. 5e), the two edges have different structures, i.e., one

armchair and the other zigzag (Mo- or S-terminated). Therefore, for a sample containing both 60° and 90° edge angles, the four edges should contain either three AC and one ZZ or one AC and three ZZ. Indeed, on TERS line imaging of the sample shown in Fig. 5d, we detected three upshifted edge and one downshifted edge, which can be convincingly assigned to AC and ZZ edges following the conclusion of our rigorous theoretical calculation, respectively. For clarity, we revised the Figure 5 and added the following discussion into the main text on Page 12:

“To further verify the conclusion drawn above, we mechanically exfoliated 1L MoS₂ with different edge angles (60° and 90°) on the Au substrate, as indicated by dot lines in Fig. 5d. These special angles would provide the structure information of the two adjacent edges^{1,2,3}. For an angle of 60° (Fig. 5e), both edges have the same structure (either zigzag or armchair). The frequency shift direction of the A_{1g} mode at both edges should be either downshift or upshift. On the other hand, for an angle of 90° (Fig. 5e), the two edges have different structures, i.e., one armchair and the other zigzag (Mo- or S-terminated). The A_{1g} frequency at these two edges should shift to different directions, i.e., one upshift and the other downshift. Therefore, for a sample containing both 60° and 90° edge angles, the four edges should contain either three AC and one ZZ or one AC and three ZZ. Indeed, on TERS spectra of these edge (Fig. 5f), we detected three upshifted edge and one downshifted edge, which can be convincingly assigned to AC and ZZ edges following the conclusion of our rigorous theoretical calculation, respectively. Therefore, we demonstrate here that TERS can conveniently identify the edge type of TMDC materials under the ambient condition, which will be very important for the practical application of TMDCs.”

Figure 5 | Effect of the edge structure on the peak position of the Raman A_{1g} mode. (a) Typical line-trace TERS spectra of the zigzag edge (left panel) and armchair edge (right panel) in the spectral range of the A_{1g} mode. Note that the spectra have been normalized with the intensity of A_{1g} peak for a better comparison. (b) Plots of peak position with the tip position. The corresponding line-trace TERS spectra are shown in Supplementary Fig. S16a and Supplementary Fig. S20a, respectively. The dash red lines are guides for the eye. (c) Calculated Raman spectra and lattice vibration of the basal plane, zigzag nanoribbon (ZNR, with a width of 3.59 nm)

localized at the Mo and S edges, and armchair nanoribbon (ANR, with a width of 2.05 nm). The envelope of each Raman spectrum is evaluated by smearing the peaks with half width 5 cm^{-1} . Note that the displacements of A_{1g} or A_{1g} -like mode with out-of-plane vibration are side viewed. See Supplementary Section 3.4 for details. **(d)** AFM image of a mechanically exfoliated 1L MoS_2 with different edge angles on a Au substrate. **(e)** Illustration of the relationship between angles and edge structures of zigzag (ZZ) and armchair (AC) in 2H MoS_2 . **(f)** TERS spectra of four edges in the spectral range of the A_{1g} mode marked in **d**. Note that the spectra have been normalized with the intensity of A_{1g} peak for a better comparison. The corresponding line-trace TERS spectra of these edges are shown in Supplementary Fig. S22.

2. If the claim of identification of edge types is discounted, the only remaining ‘discovery’ is the observation of the 396 cm^{-1} peak at the edge of bilayer MoS_2 . I am not sure if it is significant enough to justify the publication of this work in this journal.

Response: As discussed in Question 1, the correlation between the edge types and TERS band shifts has been successfully established, which further validates our conclusion. Here, we would like to further emphasize the central novelty of our work is that we not only observed a new electron-phonon interaction related Raman peak at 396 cm^{-1} in the defect of bilayer MoS_2 , but also successfully determined the lengths of electronic transition regions induced by different edge defects. Furthermore, we demonstrated that TERS can be effectively used to identify edge structures. This work would assist in revealing the structure-function relationship of the defect and subsequently guide the effective defect engineering and promote the applications of TMDC materials. Therefore, we believe this work is novel enough to match the standard of *Nature communications*.

3. Minor point. The colors of the vertical dashed lines in Fig. 3c are reversed (red vs. blue). They were correct in the previous version.

Response: Thanks for pointing out the problem and we have revised this figure.

1. You Y, Ni Z, Yu T, Shen Z. Edge chirality determination of graphene by Raman spectroscopy. *Appl. Phys. Lett.* **93**, 163112 (2008).
2. Tinoco M, Maduro L, Masaki M, Okunishi E, Conesa-Boj S. Strain-dependent edge structures in MoS₂ layers. *Nano Lett.* **17**, 7021-7026 (2017).
3. Guo Y, *et al.* Distinctive in-plane cleavage behaviors of two-dimensional layered materials. *ACS Nano* **10**, 8980-8988 (2016).

Reviewers' Comments:

Reviewer #3:

Remarks to the Author:

Although it would have been nice to have direct experimental confirmation of the edge types, the authors' argument is reasonable enough. I hope that follow-up studies will provide more direct confirmation of the authors' conclusions. With this issue resolved for the time being, I can recommend the publication of this work in Nature Comm.